# The Dps4 from *Nostoc punctiforme* ATCC 29133 is a member of His-type FOC containing Dps protein class that can be broadly found among cyanobacteria

Christoph Howe[1], Vamsi K. Moparthi[1¤], Felix M. Ho[1], Karina Persson[2]*, Karin Stensjö[1]*

**1** Department of Chemistry-Ångström Laboratory, Uppsala University, Uppsala, Sweden, **2** Department of Chemistry, Umeå University, Umeå, Sweden

¤ Current address: Department of Physics, Chemistry and Biology, Division of Chemistry, Linköping University, Linköping, Sweden
* karin.stensjo@kemi.uu.se (KS); karina.persson@umu.se (KP)

**Data Availability Statement:** All relevant data are within the manuscript and its Supporting Information files.

## Abstract

Dps proteins (DNA-binding proteins from starved cells) have been found to detoxify $H_2O_2$. At their catalytic centers, the ferroxidase center (FOC), Dps proteins utilize $Fe^{2+}$ to reduce $H_2O_2$ and therefore play an essential role in the protection against oxidative stress and maintaining iron homeostasis. Whereas most bacteria accommodate one or two Dps, there are five different Dps proteins in *Nostoc punctiforme*, a phototrophic and filamentous cyanobacterium. This uncommonly high number of Dps proteins implies a sophisticated machinery for maintaining complex iron homeostasis and for protection against oxidative stress. Functional analyses and structural information on cyanobacterial Dps proteins are rare, but essential for understanding the function of each of the NpDps proteins. In this study, we present the crystal structure of NpDps4 in its metal-free, iron- and zinc-bound forms. The FOC coordinates either two iron atoms or one zinc atom. Spectroscopic analyses revealed that NpDps4 could oxidize $Fe^{2+}$ utilizing $O_2$, but no evidence for its use of the oxidant $H_2O_2$ could be found. We identified $Zn^{2+}$ to be an effective inhibitor of the $O_2$-mediated $Fe^{2+}$ oxidation in NpDps4. NpDps4 exhibits a FOC that is very different from canonical Dps, but structurally similar to the atypical one from DpsA of *Thermosynechococcus elongatus*. Sequence comparisons among Dps protein homologs to NpDps4 within the cyanobacterial phylum led us to classify a novel FOC class: the His-type FOC. The features of this special FOC have not been identified in Dps proteins from other bacterial phyla and it might be unique to cyanobacterial Dps proteins.

## Introduction

Dps proteins (DNA-binding proteins from starved cells), also referred to as miniferritins, are only found in prokaryotes and belong to the class of ferritin-like proteins [1], alongside with bacterioferritins (bfr) and ferritins (ftn). Dps proteins exhibit a remarkable three-dimensional

**Funding:** This work was supported by KS: NordForsk (project # 82845) (https://www. nordforsk.org/en), the NCoE program "NordAqua", KS: Swedish Energy Agency (https://www. energimyndigheten.se/en/), (project # 11674-5); VKM: Post Doc scholarship, Sven and Lilly Lawski foundation, CH: Stiftelsen Lars Hiertas Minne. The funders had no role in study design, data collection and analysis, decision to publish, or preparation of the manuscript.

**Competing interests:** The authors have declared that no competing interests exist.

structure consisting of twelve monomers (or six dimers), forming a spherically shaped protein complex with a hollow spherical interior [2,3].

On the inside, each dimeric interface forms two identical catalytic centers, called the ferroxidase centers (FOC). There, the oxidation of ferrous iron ($Fe^{2+}$) to ferric iron ($Fe^{3+}$) takes place and an iron oxide mineral core consisting of up to 500 Fe atoms can be formed [4]. Canonical FOCs in Dps proteins consist of five conserved amino acids, namely two His and one Asp from one monomer as well as one Glu and one Asp from the adjacent monomer at the dimer interface.

To reach the catalytic center, the $Fe^{2+}$ ions have been suggested to travel through two types of pores that are connecting the internal cavity with the exterior [2]. One pore type is the ferritin-like pore, which is named after their structural similarity to the iron entrance pores in ferritins. The ferritin-like pore has been frequently assigned to be the iron entrance pore due to its negatively charged character in canonical Dps structures. The other pore type, the Dps-type pore, is specific to Dps proteins. Compared to the ferritin-like pore it has a more hydrophobic character and its function remains unknown [5].

When $Fe^{2+}$ has entered the Dps and bound to the FOC, it is usually oxidized by $O_2$ or $H_2O_2$. However in canonical Dps proteins, $O_2$ is utilized at a 100-fold lower reaction rate as compared to $H_2O_2$, as demonstrated for *Escherichia coli* Dps (EcDps) [4,6]. Due to the consumption of $H_2O_2$ and $Fe^{2+}$ a mineral core is formed and the Fenton reaction is circumvented, from which in the absence of the Dps protein highly toxic hydroxyl radicals are produced [7].

Thus the Dps protein class joins the cellular protection machinery alongside catalases and peroxidases that disarm reactive oxygen species (ROS) [8,9]. Cyanobacteria, which contain the $O_2$-evolving photosynthetic apparatus, require an effective response to ROS, as $O_2$ is the precursor for all ROS including $H_2O_2$ [10,11]. The filamentous diazotrophic cyanobacterium *Nostoc punctiforme* ATTC 29133 exhibits a complex $H_2O_2$-protection system including three catalases and various peroxidases [12]. In addition to those, we have identified five Dps proteins [13].

In cyanobacteria the existence of multiple Dps proteins has been found to be rather common [13,14], but most of the more well-studied Dps proteins originate from pathogens that possess only one or two Dps proteins [2,15–17]. Information about cyanobacterial Dps proteins and their involvement in iron homeostasis and ROS defense is scarce. To learn specifically more about cyanobacterial Dps proteins, we have collected data on the different NpDps from *N. punctiforme* in the past years to identify their physiological roles. In an earlier study [13] we hypothesized that, despite all belonging to the Dps protein family, they might differ in their biochemical properties and thus in their physiological function.

First we have identified that, with the exception of *Npdps2*, all the *Npdps* genes are predominantly transcribed in heterocysts [18], a specific cell type that is developed in a regular pattern along the filament during nitrogen depletion. In heterocysts a microoxic environment is created, necessitated by the oxygen sensitivity of the nitrogenase that is involved in $N_2$-fixation [19]. In an earlier *in vivo* study we additionally observed that NpDps2 (Npun_F3730) is an essential protectant against $H_2O_2$ that is exogenously added to the cyanobacterial culture [13]. Interestingly, biochemical investigations showed that NpDps1, NpDps2 and NpDps3 could catalyze the oxidation reaction of $Fe^{2+}$ to $Fe^{3+}$ with $H_2O_2$, but not with $O_2$ [20]. NpDps1-3 showed high sequence similarities with the canonical Dps [20]. *In silico* experiments indicated that NpDps1-3 likely import $Fe^{2+}$ through their ferritin-like pores as commonly hypothesized for typical Dps proteins [20]. Additionally we discovered that NpDps2 and NpDps5 (Npun_F6212), are involved in the tolerance against high light intensities [21], a condition that enhances the production of ROS [22,23].

The information about NpDps4 is limited and its role in *N. punctiforme* is not resolved [13]. NpDps4 shares a sequence identity of circa 64% (based on the sequences derived by Cyanobase [24] and alignment by Clustal Omega [25]) with the DpsA from the cyanobacterium *Thermosynechoccus elongatus* (TeDpsA). Notably, *T. elongatus* exhibits two Dps proteins, which are the only cyanobacterial Dps proteins that have been structurally studied so far. While TeDps was found to share high similarities with canonical Dps [5], TeDpsA was found to be an atypical Dps. TeDpsA has been reported to exhibit surprisingly similar $Fe^{2+}$ oxidation rates with $H_2O_2$ and $O_2$, which was suggested to relate to $Zn^{2+}$ bound at the FOC [14]. Canonical FOCs are usually dominated by negatively charged carboxyl groups from various Glu and Asp among two conserved His [14]. Instead, the FOC of TeDpsA was found to have a non-canonical nature with two additional His and less involvement of carboxyl groups from Asp or Glu. Our sequence alignment revealed that NpDps4 may exhibit a similar FOC as the one from TeDpsA [13]. Whether this type of FOC might be found among other cyanobacterial Dps has not yet been resolved.

In this study we investigate the structural and biochemical properties of NpDps4 to gain insights on its physiological role in *N. punctiforme*. We gathered crystal structure data on NpDps4 in a metal-free, an iron-bound and zinc-bound form. Additionally, the purified protein was analyzed in a series of spectroscopic analyses to identify whether $O_2$ or $H_2O_2$ could be used for $Fe^{2+}$ oxidation under aerobic conditions. We further studied whether we could observe a similar effect of $Zn^{2+}$ on the enzymatic reaction as compared to TeDpsA. Finally, we reviewed our earlier phylogenetical data about Dps proteins, giving rise to the suggestion of a new classification of ferroxidase centers and a potentially new class of Dps proteins.

## Results

### Structural compartments of the NpDps4 dodecamer and insight on metal ion trafficking

Dps crystal structures from more than 30 different bacteria have been determined since the Dps protein family was discovered in 1992 [26]. Only two Dps structures originate from the cyanobacterial phylum [5,14], while the major focus in the field has been on Dps from pathogens. In the following we show the crystal structure of NpDps4 from the cyanobacterium *N. punctiforme*. The structural data gives insights into the overall multimerization, the pore structures and the FOC. We soaked the crystals in different metal ion solutions to identify possible metal binding sites. Such information can give insights on possible metal ion co-factors or inhibitors of the protein.

The overall structure of NpDps4 forms a dodecameric hollow sphere that is built up from identical monomers (Fig 1A). Each monomer consists of 184 amino acids that form a four-helix bundle consisting of the helices A, B, C and D (Fig 1B). A small BC helix is integrated within the long loop that connects helix B and C. An N-terminal tail of 22 amino acids precedes helix A, and a C-terminal tail of 12 amino acids follows helix D. As twelve monomers build up the dodecamer, dimeric as well as trimeric sub-structures can be observed (Fig 1A). The dimer is formed by helices A and B from one monomer packing head-to-tail to the corresponding helices from a second monomer. In addition, the short BC helices from the two monomers run anti-parallel to each other and interact mainly by hydrophobic interactions promoted by the side chains of Phe95, Leu98 and Ala99 (S1 Fig).

A pore is present at the trimeric interface, mainly formed by the short loops that link helices C and D (Fig 2). The formation of such a pore is a well-known feature of Dps proteins and is called a ferritin-like pore, since it is formed along a 3-fold axis, similar to the pores in ferritins [27]. In the case of NpDps4, charged and aromatic residues from the three monomers line the

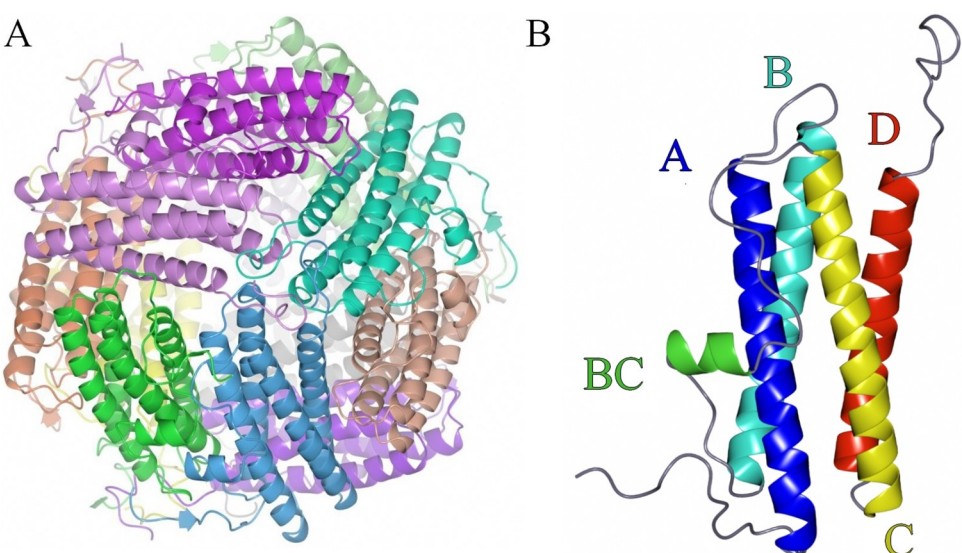

**Fig 1. Overall structure of NpDps4.** A, view on the dodecameric sphere, where the subunits are shown in different colors and B, the Dps4 monomer, helices (A, B, BC, C and D) indicated with individual colors and color-corresponding letters.

ferritin-like pore. Each subunit of the trimer contributes symmetrical side chains of Glu140 facing the exterior of the pore, followed by Arg145, Arg148 and Glu152, which all form a network of hydrogen bonds. Tyr149 as well as Lys153 face the hollow interior of the Dps protein. There is a tight hydrogen bonding network between the residues that line the pore, leaving no room for bound water molecules or ions.

Another pore is formed by the short loops between helix A and B of three monomers (Fig 3). This pore is described as the Dps-type pore as it is unique for Dps proteins within the

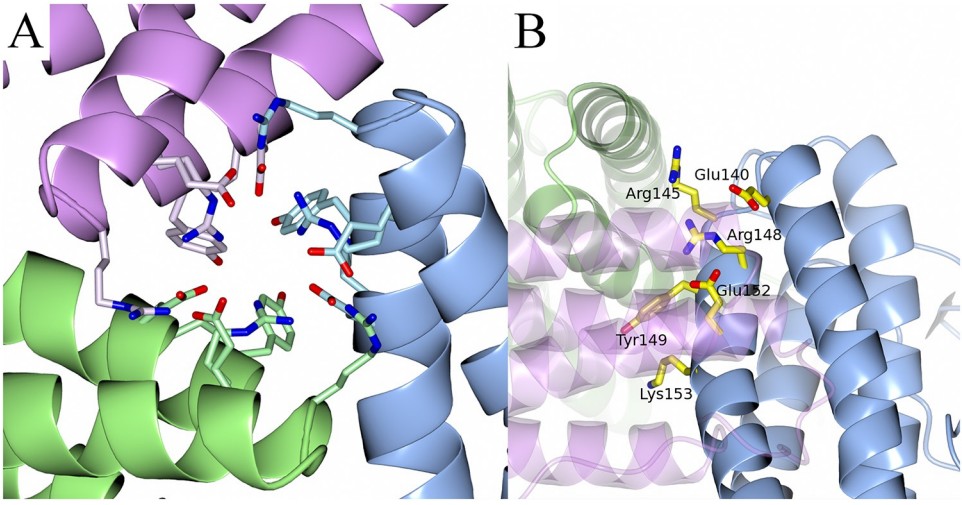

**Fig 2. The ferritin-like pore of NpDps4.** A. View along the 3-fold axis at the trimeric ferritin-like pore interface with amino acid side chains lining the pore as stick models and monomers displayed as ribbon structure. Side chains are colored as their respective monomer. B. Side view on the 3-fold axis with the side chains of one monomer lining the pore are depicted as stick models.

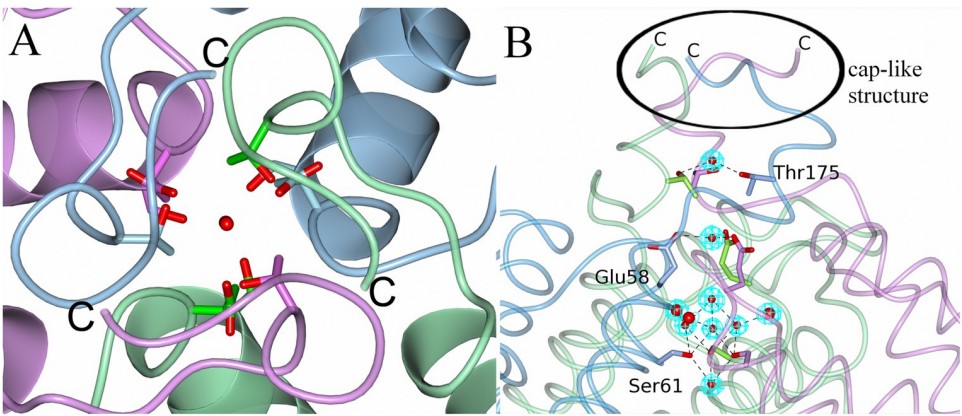

**Fig 3. The Dps-type pore of the NpDps4.** A. View along the 3-fold axis at the trimeric C-terminal interface with amino acid side chains lining the pore as stick models. The subunits are depicted in pink, blue and green. B. Perpendicular perspective of the 3-fold axis with amino acid side chain lining the pore as stick models and monomers colored as in A. Each symmetrical amino acid trio is displayed, but labelled only once. C-termini indicated with the letter 'C'. The cap-like structure on top of the pore is highlighted. The water molecules are illustrated as red spheres in a 2Fo-Fc electron density map contoured at 1σ.

ferritin-like protein family [28]. Each trimeric subunit contributes three symmetrical side chains of Thr175 facing the protein exterior, followed by Glu58 and Ser61 that are lined up along the pore and form hydrogen bonds with several water molecules inside the pore (Fig 3B). The pore has a polar chemical character and is capped by the three C-terminal tails (sequence FVQAA) on the outside of the protein (Fig 3B).

The NpDps4 structure was determined in three forms; metal-free, Fe-soaked and Zn-soaked. Except for the metal binding sites the three structures were very similar. A pairwise comparison between the equivalent subunits of the three proteins shows that root mean square deviation values (r. m. s. d.) vary between 0.16 and 0.23 Å. Due to the higher resolution of the metal-free NpDps4 more amino acids could be modelled in the N-terminus of each subunit: two additional residues in three of the subunits and eight in one subunit (including two from the linker). In addition, a double conformation of Cys103 is observed in metal-free Dps4 but not in the metal bound forms.

## Iron and zinc bind to the ferroxidase center

In the metal-free crystals, which diffracted to 1.6 Å, no remnants of metal atoms were detected. This is presumably due to the chelating capacity of the crystallization agent SOKALAN HP66 (a vinylpyrrolidone/vinylimidazole polymer). However, when crystals were transferred to a polyethylene glycol (PEG) solution containing $Zn^{2+}$ or $Fe^{2+}$ ions, these metals were observed in the structure (Fig 4 and S2 Fig).

In the $Fe^{2+}$-soaked crystals, which diffracted to 1.9 Å, two strong electron density peaks were found on the dimeric interface at the FOC (S2 Fig). According to canonical Dps nomenclature, usually two sites (site A and B) within the FOC have been assigned. The Fe atom at the A-site is coordinated by His78 and Glu82 from helix A from one monomer and His51 from helix B from a neighboring monomer (Fig 4A). The Fe atom in site B is located only 3.5 Å from the Fe atom in site A and is coordinated by the same Glu82 from helix A, the His63 from helix B, and the His164 from a helix of a third monomer at coordination distance. In canonical FOCs usually a conserved negatively charged Asp or Glu at position 67 has been shown to participate in Fe coordination at site B [29–31]. In NpDps4, the corresponding Asn67 is not

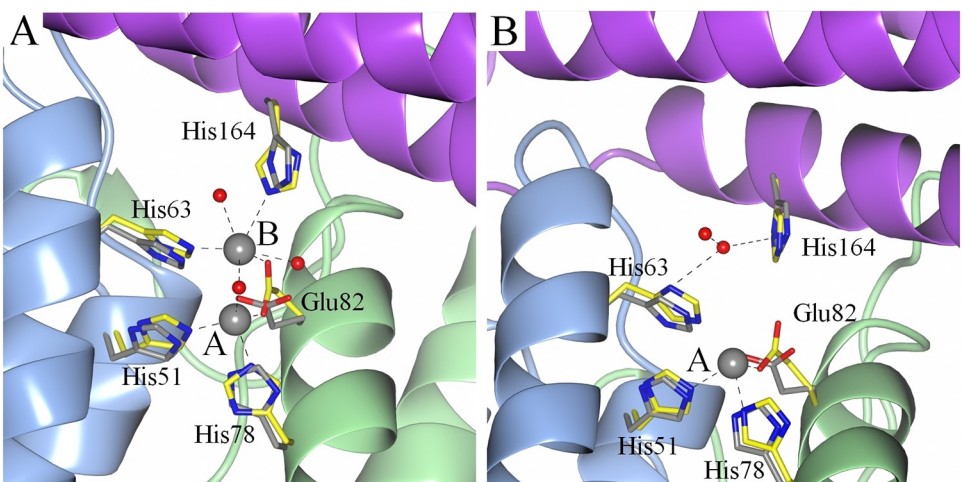

**Fig 4. Superposition of the amino acids involved in the formation of the NpDps4 ferroxidase center.** A. The $Fe^{2+}$-soaked structure (amino acids colored in yellow, metal coordination indicated with dashed line) with the metal-free structure (amino acids colored in grey). B. The $Zn^{2+}$-soaked structure (amino acids colored in yellow, metal coordination indicated with dashed line) with the metal-free structure (amino acids colored in grey). Red spheres indicate water molecules and grey spheres indicate the Fe atoms (A) and the Zn atom (B). The metal coordination sites of the FOC are indicated with A and B.

participating in FOC formation, neither the adjacent Asp68 (not shown). Among different water molecules that were found in the coordination sphere of the Fe atoms one was found to coordinate to both Fe atoms at site A and B at distances of 2.2 Å and 2.3 Å, respectively (S1 Table). The water molecule is not located on a straight line between the Fe atoms. The angle between the two Fe atoms and the bridging water molecule is 95°. When the $Fe^{2+}$-soaked structure was compared to the metal-free structure, a pronounced conformational change was observed. Both, Glu82 and His164 were tilted towards the metal atoms for better coordination (Fig 4A).

In the $Zn^{2+}$-soaked crystals, which diffracted to 2.4 Å, the A-site was occupied with a Zn atom (Fig 4B). The Zn atom was coordinated by His78 and Glu82 from helix A and His51 from helix B, which were the same coordinating amino acids as for the Fe atom in site A. Also, here a pronounced conformational change can be observed for Glu82 for better coordination to the Zn atom. No conclusive information about a second metal could be derived for site B due to the limited resolution, however at some of the B-sites a low occupancy Zn atom could be indicated based on anomalous electron density maps (S2 Fig.). Additional metal binding sites for zinc or iron were not observed in the overall crystal, but cannot be excluded. All the coordination distances in the A and B sites are presented (S1 Table).

## NpDps4 utilizes $O_2$ to oxidize $Fe^{2+}$

Dps proteins catalyze the oxidation reaction of $Fe^{2+}$ to $Fe^{3+}$ with $O_2$ and $H_2O_2$, thereby forming an iron mineral core inside the protein cavity. The oxidant affinity towards $H_2O_2$ is up to 100-fold greater as compared to $O_2$ [4,6]. The ability to consume $H_2O_2$ during the oxidation has been shown to be of importance for cellular defenses against oxidative stress [13,32,33]. Our earlier results on the NpDps1-3 showed that none of these three proteins could catalyze the $O_2$-mediated oxidation reaction, while $H_2O_2$ acted as a good oxidant [20].

To investigate whether NpDps4 exhibits similar oxidant preferences as the canonical NpDps we performed a qualitative absorption spectroscopy analysis in which oxidation of

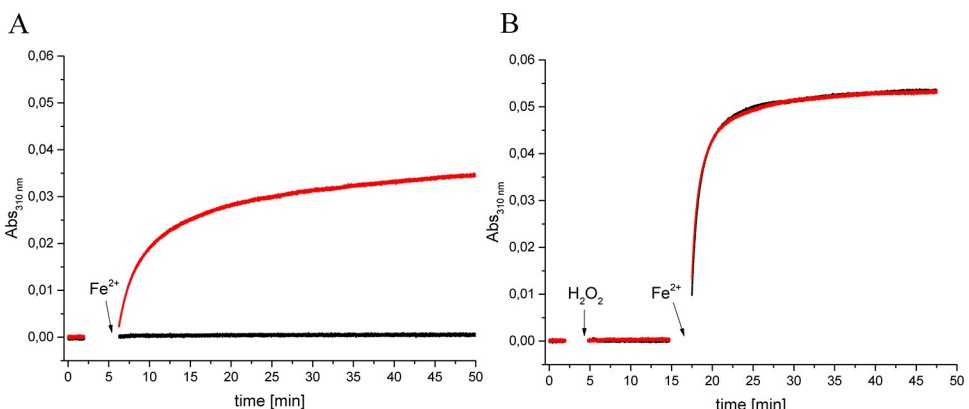

**Fig 5. NpDps4 catalyzes the $O_2$-mediated $Fe^{2+}$ oxidation, but no indication for utilizing $H_2O_2$ as an oxidant was found.** Absorption spectroscopy following the wavelength at 310 nm to monitor the formation of $Fe^{3+}$ species over time. A. In the absence of NpDps4, 24 μM $Fe^{2+}$ was added to the reaction mixture (black graph). In the presence of 0.5 μM NpDps, 24 μM $Fe^{2+}$ was added to the reaction mixture (red graph). B. In the absence of the protein 16 μM $H_2O_2$ and 24 μM $Fe^{2+}$ were added (black graph). In presence of 0.5 μM NpDps4, 16 μM $H_2O_2$ and 24 μM $Fe^{2+}$ were added (red graph). The reaction mixture contained 5 mM succinate buffer at pH 6.0 and 50 mM NaCl under aerobic conditions. Protein presence is indicated (red graph). All chemical additions are highlighted with arrows. Protein absorbance was set to zero. Time axis was reconstructed from subsequent absorbance traces, because spectra were recorded after each chemical addition. Two experimental replicates were performed for each reaction; the same NpDps4 protein material as for metal-free crystal structure determination was utilized (S1 File).

$Fe^{2+}$ to $Fe^{3+}$ was determined. While $Fe^{2+}$ is transparent at the wavelength 310 nm, the formation of $Fe^{3+}$ is accompanied with a clear absorbance increase at this wavelength (Fig 5) [4,34]. All experiments were done under aerobic conditions. During an incubation time of 2 min the absorbance at 310 nm did not increase in the reaction buffer (5 mM succinate buffer at pH 6.0 and 50 mM NaCl) in the presence of 0.5 μM NpDps4 (Fig 5A, red curve). When 24 μM $Fe^{2+}$ was added, the absorbance increased logarithmically with a fast initial increase reaching $Abs_{310nm}$ 0.014 after 2 min, which represented ~ 26% of the maximum absorbance (max. $Abs_{310nm}$ = 0.053) in this reaction. In the following 30 min the reaction slowed down and the $Abs_{310nm}$ reached 0.032 and was monitored further to a total of 420 min reaching an absorbance of 0.046 (S1 File). At a ratio of 48 $Fe^{2+}$/Dps protein all $Fe^{2+}$ binding sites inside the Dps dodecamer could be theoretically occupied, as there are 12 FOC with each two iron binding sites per Dps protein (24 $Fe^{2+}$/dodecamer binding sites). In the control experiment, that contained no protein material, 24 μM of $Fe^{2+}$ was added to the reaction mixture, but no change in the absorbance was detected for the next 45 min (Fig 5A, black curve), neither for during the following 420 min (S1 File). Under the control condition no iron oxide was formed. We verified that $O_2$ was the necessary oxidant for the rise in absorbance at 310 nm in the presence of NpDps4 (S3 Fig). The increase in absorbance at 310 nm could be stopped with the establishment of anaerobic conditions (sparging with argon), while subsequent aeration would lead to the absorbance rise to recommence (S3 Fig).

To investigate whether NpDps4 could utilize $H_2O_2$ as an oxidant in the $Fe^{2+}$ oxidation reaction we used the same spectroscopic approach, monitoring the absorbance at 310 nm to follow $Fe^{2+}$ oxidation over time. All experiments were done under aerobic conditions. In the presence of 0.5 μM NpDps4 during the incubation time of 2 min the signal at 310 nm did not change in the reaction buffer (Fig 5B, black curve). Then 16 μM $H_2O_2$ was added to the reaction mixture and monitored for 10 min, but no change in the absorbance was detected. Subsequently, 24 μM $Fe^{2+}$ was added to the reaction mixture and the absorbance increased logarithmically with a fast initial increase reaching an $Abs_{310nm}$ of 0.040 after 2 min. Thereafter the absorbance

reached its maximal value of approx. 0.053 after 20 min and from there on kept unchanged. The oxidation of $Fe^{2+}$ by $H_2O_2$ was also observed, in the absence of NpDps4 (Fig 5B, red curve). Thus no additional effect as a result of the presence of NpDps4 could be observed for the oxidation of $Fe^{2+}$ by $H_2O_2$, or was too small and therefore not detected. An effect of the $O_2$-mediated $Fe^{2+}$ oxidation reaction catalysed by NpDp4 was not observed to contribute to the increase of absorbance at 310 nm in the presence of $H_2O_2$.

## $Zn^{2+}$ is a potent inhibitor of NpDps4

Zinc has been found to bind at the FOC in ferritin-like proteins. Generally zinc was found to competitively inhibit the $O_2$-mediated oxidation reaction of $Fe^{2+}$ to $Fe^{3+}$ in Dps proteins [35–37] as well as in maxiferritins (ftn and bfr) [38–40]. To investigate what effect Zn binding has on the $Fe^{2+}$ oxidation activity of NpDps4, we performed spectroscopic experiments following the $Fe^{2+}$ oxidation at 310 nm over time. When 24 μM $Fe^{2+}$ was added to the reaction mixture containing 0.5 μM NpDps4, pre-incubated with 12 μM $Zn^{2+}$, no change in the absorbance was detected (Fig 6A, red graph). In the the control reaction, no pre-incubation with $Zn^{2+}$, $Fe^{2+}$ was oxidized as shown as an increase in absorbance (Fig 6A, black graph), as previously described (Fig 5A). We concluded that zinc could effectively inhibit the oxidation reaction under aerobic conditions. Additionally, an increase in background absorbance between 200–800 nm was detected when $Zn^{2+}$ and NpDps4 were simultaneously present (S1 File). We concluded that $Zn^{2+}$ negatively affected the solubility of NpDps4 leading to slow protein precipitation or aggregation.

To resolve if the precipitation/aggregation of the NpDps4 in presence of $Zn^{2+}$ was the origin of the enzymatic inhibition, we investigated whether zinc could directly halt an ongoing oxidation reaction. 24 μM $Fe^{2+}$ was added to 0.5 μM NpDps4 under aerobic conditions and $Fe^{2+}$ was oxidized (rise in absorbance at 310 nm). After 5 min, 12 μM $Zn^{2+}$ was added and the rise in absorbance arrested within the next 30 seconds (Fig 6B, red graph), Such a sudden reaction stop was not detected in the control experiment, in which the reaction between 24 μM $Fe^{2+}$ and 0.5 μM NpDps4 resulted in a steady increase of the absorbance over time (Fig 6B, black graph). The addition of $Fe^{2+}$ and $Zn^{2+}$ to the reaction mixture in the absence of protein did not alter the absorbance over time (S1 File).

## Occurrence of homologous structures of NpDps4 across the cyanobacterial phylum

Since the discovery of the Dps protein family in 1992 [26], Dps proteins from pathogens have been the main focus of biochemical investigation. Among the investigated Dps proteins, sequence alignments and crystal structure information revealed the existence of a coherent structural compartment, the canonical FOC. In an earlier study we discovered that NpDps4 might structurally differ from the canonical Dps [13]. By sequence comparison we identified the His78 replacing the highly conserved Asp typically for the canonical FOC. In this study we verified via crystal structure analysis that the His78 was indeed involved in the coordination of iron in FOC at site A. As this amino acid exchange has not been demonstrated for any of the non-cyanobacterial Dps proteins studied so far, we asked the question whether the His78 could be found among other cyanobacterial Dps proteins besides the TeDpsA that contains it [14]. Here we present an alignment of cyanobacterial protein sequences that are homologous to NpDps4 (Fig 7) [41]. For comparison we included canonical Dps sequences from *Listeria innocua* and *Escherichia coli*, but also cyanobacterial ones from *N. punctiforme* and *T. elongatus* that were earlier found to cluster with canonical Dps [5,13,20]. The fifth Dps of *N. puntiforme* (NpDps5) was not included in the sequence alignment as it has been shown to

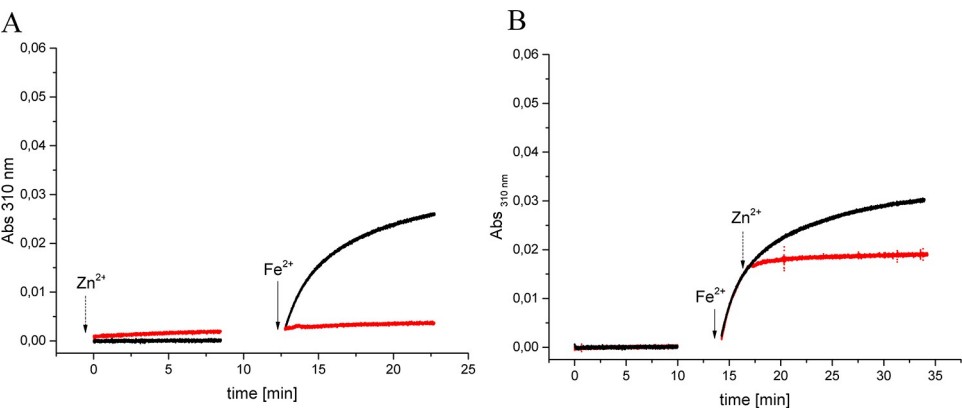

**Fig 6. Zn²⁺ inhibits NpDps4 catalyzing the O₂-mediated Fe²⁺ oxidation.** Absorption spectroscopy following the wavelength at 310 nm to monitor the formation of $Fe^{3+}$ species [4,34]. A. In the presence of 0.5 μM NpDps incubated with 12 μM $Zn^{2+}$ for 8 min (red graph), or without the addition of 12 μM $Zn^{2+}$ (black graph), 24 μM $Fe^{2+}$ was added to the reaction mixture (solid-lined arrow). B (red graph) In the presence of 0.5 μM NpDps, 24 μM $Fe^{2+}$ was added (solid-lined arrow) and after 5 min 12 μM $Zn^{2+}$ was added (dashed arrow) to the reaction mixture. B (black graph) In the presence of 0.5 μM NpDps4 24 μM $Fe^{2+}$ was added (solid-lined arrow) to the reaction mixture (black graph) and no $Zn^{2+}$ was added. The reaction mixture contained 5 mM succinate buffer at pH 6.0 and 50 mM NaCl under aerobic conditions. All chemical additions are highlighted with arrows. Protein absorbance was set to zero. Time axis was reconstructed from subsequent absorbance traces, because spectra were recorded after each chemical addition. Two experimental replicates were performed for each reaction; the same NpDps4 protein material was utilized as for metal-free crystal structure determination (S1 File).

comprise a FOC ligand sphere very different to that of canonical Dps and similar to that of Bfrs [13].

We found that the His78, with some exceptions, can be broadly found among cyanobacterial Dps sequences. In strains within five of the six phylogenetic groups [41], namely in Nx (*Nostocales sensu lato* + others), LPP-B (*Leptolyngbya + Nodosilinea + Synechococcus*), AcTh (*Acaryochloris + Thermosynechococcus*), Osc (*Oscillatoriales sensu stricto*) and SPM (*Synechocystis + Pleurocapsa + Microcystis*) we found Dps protein sequences that possessed the His78. Some of the aligned Dps sequences within the groups of Nx (Cal6303_4560) and SPM (SynPCC7002_A0031, Cyan10605_1025, Slr1894, MAE_62840) did not contain the His78, but instead the conserved Asp at the equivalent position. A further analysis of the genomic data (http://genome.microbedb.jp/cyanobase/) showed that other putative Dps proteins within the investigated strains exhibited only canonical FOC. All Dps homolog sequences to NpDps4 that clustered within the phylogenetic group of SynPro (*Syncechococcus + Prochlorococcus + Cyanobium*) and *Gloeobacter* species were not found to exhibit the His78, but the conserved Asp or Glu as found in canonical FOCs (Fig 7). Also the analysis of the genomic data (http://genome.microbedb.jp/cyanobase/) showed that all putative Dps within the investigated strains contained canonical FOCs. Only a limited number of sequenced cyanobacterial genomes are available and a selection of representative Dps homologs from different phylogenetic groups was utilized in this study. Therefore, it cannot be excluded that the His78 might be found in Dps sequences from other strains within the SynPro group. Nevertheless, we can conclude that the FOC motif containing the His78 is broadly spread across the cyanobacterial phylum including unicellular, filamentous, filamentous heterocystous cyanobacteria from both, marine and freshwater environments.

Interestingly, the His164 that coordinates to the iron at FOC site B in NpDps4 is frequently found among the aligned Dps sequences cyanobacterial Dps that simultaneously possess the His78. This correlation was not seen for the cyanobacterium *Gloeobacter violaceus* PCC 7421

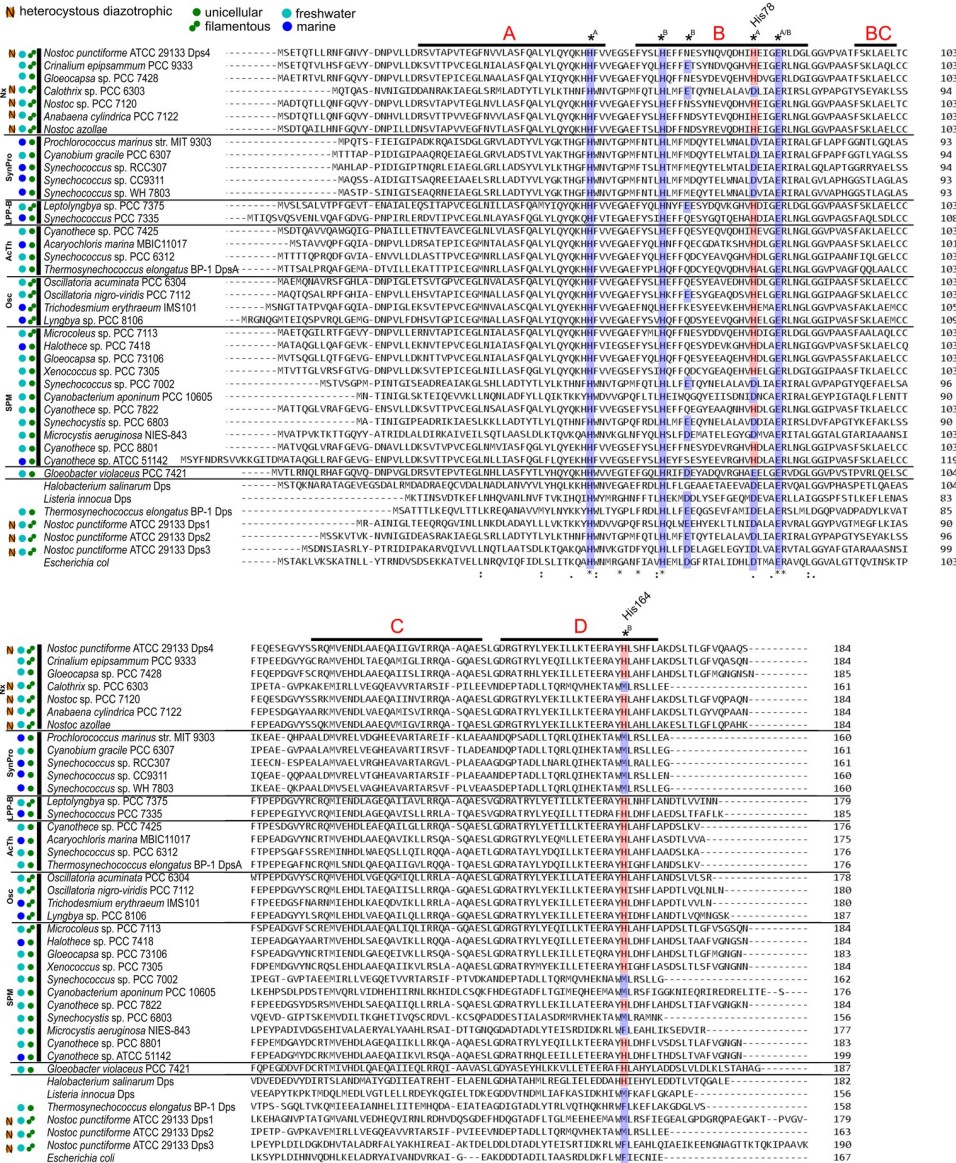

**Fig 7. Multiple sequence alignment of the NpDps4 with a selection of its homolog sequences from six cyanobacterial phylogenetic groups.** The black bar on top of the NpDps4 sequence represents the α-helices found in the NpDps4 crystal structure and the red letter (A,B,BC,C and D) refer to their helix annotation. Amino acids of the ferroxidase center (FOC) are indicated by an asterisk *A, *B or *A/B indicating the involvement of metal coordination at site A and/or B in regards to the His-type FOC of NpDps4 and TeDpsA. Conserved amino acids shared by canonical FOCs are highlighted in blue. Non-canonical His78 and His164 are highlighted in red. Cyanobacterial sequences were chosen to represent members of recent phylogenetic classification among cyanobacteria [41], namely Nx (*Nostocales sensu lato* + others), LPP-B (*Leptolyngbya* + *Nodosilinea* + *Synechococcus*), AcTh (*Acaryochloris* + *Thermosynechococcus*), Osc (*Oscillatoriales sensu stricto*), SPM (*Synechocystis* + *Pleurocapsa* + *Microcystis*) and SynPro (*Syncechococcus* + *Prochlorococcus* + *Cyanobium*). The alignment was conducted with sequences of Cri9333_4135, *Crinalium epipsammum* PCC 9333; Glo7428_4653, *Gloeocapsa* sp. PCC 7428; Cal6303_4560, *Calothrix* sp. PCC 6303; alr3808, *Nostoc* sp. PCC 7120; Anacy_2689, *Anabaena cylindrica* PCC 7122; Aazo_0570, *Nostoc azollae* 0708; P9303_29571, *Prochlorococcus marinus* str. MIT 9303; Cyagr, *Cyanobium gracile* PCC 6307; SynRCC307_2440, *Synechococcus* sp. RCC307; sync_2856, *Synechococcus* sp. CC9311; SynWH7803_2460, *Synechococcus* sp. WH 7803; Lepto7375DRAFT_1349, *Leptolyngbya* sp. PCC 7375; S7335_1356, *Synechococcus* PCC 7335; Cyan7425_4372, *Cyanothece* sp. PCC 7425; AM1_3409, *Acaryochloris marina* MBIC11017; Syn6312_2228, *Synechococcus* sp. PCC 6312; TeDpsA, *Thermosynechococcus elongatus* BP-1; Oscil6304_1209, *Oscillatoria acuminata* PCC 6304; Osc7112_5725, *Oscillatoria nigro-viridis* PCC 7112; Tery_4282, *Trichodesmium erythraeum* IMS101; L8106_02877, *Lyngbya* sp. PCC 8106; Mic7113_6129, *Microcoleus* sp. PCC 7113; PCC7418_0935, *Halothece* sp. PCC 7418; GLO73106DRAFT_00020910, *Gloeocapsa* sp. PCC 73106; Xen7305DRAFT_00027340, *Xenococcus* sp. PCC

7305; SYNPCC7002_A0031, *Synechococcus* sp. PCC 7002; Cyan10605_1025, *Cyanobacterium aponinum* PCC 10605; Cyan7822_3527, *Cyanothece* sp. PCC 7822; slr1894, *Synechocystis* sp. PCC 6803; MAE_62840, *Microcystis aeruginosa* NIES-843; PCC8801_2450, *Cyanothece* sp. PCC 8801; cce_0479, *Cyanothece* sp. ATCC 51142; gll0337, *Gloeobacter violaceus* PCC 7421; HsDpsA, *Halobacterium salinarum*; LiDps, *Listeria innocua*; TeDps, *Thermosynechococcus elongatus* BP-1; NpDps1-3, *Nostoc punctiforme* ATCC 29133; EcDps, *Escherichia coli*. The sequences were annotated according to Cyanobase (http://genome.microbedb.jp/cyanobase/). Information about the morphology of the cyanobacteria (unicellular or filamentous), origin of isolation (marine or freshwater) and capacity of forming heterocysts for N$_2$-fixation is depicted [41].

(gll0337) and the DpsA from *Halobacterium salinarum* (HsDpsA), an archaeal Dps. Canonical Dps usually possess a Met or Phe, at the corresponding position of His164. These amino acids are not involved in FOC formation [13,29–31,42–44].

## Discussion

In this study we have gathered structural and biochemical data that supports our previously proposed hypothesis that the NpDps4 is an atypical Dps protein. NpDps4 diverges from canonical Dps especially by its amino acid composition at the FOC. Also, no evidence for NpDps4 using H$_2$O$_2$ as an oxidant in the presence of O$_2$ was discovered.

### The His-type FOC–a novel subclass of FOC

By crystal structure analysis, we have found that the FOC in NpDps4 exhibits special features that differentiate it from the group of canonical FOCs in Dps proteins [13]. In NpDps4: (i) the His78 replaces a highly conserved Asp at the metal binding site A and is likewise involved in metal coordination, (ii) the amino acid at position 67 is an Asn that does not contribute to metal binding. This is different from the canonical FOC, in which a conserved Glu or Asp at position 67 is involved in coordination of the metal in FOC site B mediated via an intermediate water molecule [29–31], (iii) His63 coordinates directly to the Fe atom in site B, which is an unusual finding, as canonical Dps proteins also possess a His at that position, but it coordinates via an intermediate water molecule [29–31], (iv) the His164 from a third monomer is involved in the metal coordination at FOC site B. The contribution from a third monomer is unusual and novel in FOC formation. In canonical Dps non-coordinating Met or Phe are conserved at the His164 position in NpDps4 [13,29–31,42–44]. All these structural properties accord with those found in the crystal structure of TeDpsA [14], the homolog to NpDps4. To sum up, in canonical FOCs metal coordination is clearly dominated by negatively charged carboxylates. Instead NpDps4 and TeDpsA share a pronounced His character, which leads us to propose a novel FOC classification, the His-type FOC, which appears to be broadly found within the cyanobacterial phylum.

### The His-type FOC–a common feature among cyanobacterial Dps sequences

In this study we show that members in five out of six recently categorized [41] cyanobacterial phylogenetic groups possess Dps proteins that contain the His-type FOC. Even though our data set represents a limited subset of sequenced genomes, we were still able to show how broadly the His-type FOC can be found across the cyanobacterial phylum. Interestingly, features of the His-type FOC have not been identified in any other bacterial phyla implying that the His-type FOC has a specific function in the cyanobacterial metabolism such as photosynthesis. But as mentioned above, not all cyanobacteria appear to possess a Dps protein with the His-type FOC. This is the case for *Synechocystis* sp. PCC 6803, *Gloeobacter* sp. and the shown cyanobacteria within the phylogenetic group SynPro (*Syncechococcus* + *Prochlorococcus* +

*Cyanobium*). One may ask why some cyanobacteria possess a Dps protein with this unique FOC and others do not. Both, marine and freshwater cyanobacteria can possess it, and cellular morphology does not appear to be decisive in this regard as members of unicellular (e.g. *Cyanothece* sp., *Gloeobacter* sp.), colony forming (e.g. *Xenococcus* sp.), filamentous (e.g. *Oscillatoria* sp.), heterocystous (e.g. *Calothrix* sp., *Nostoc* sp.) and branching filamentous (e.g. *Hapalosiphon* sp., not shown in this study) cyanobacteria exhibit Dps containing the His-type FOC. Also, the organism's complexity in terms of cellular differentiation does not seem be a factor for the possession of this special Dps protein. This can be exemplified by *N. punctiforme* and *T. elongatus*, both of which possess the His-type FOC. *N. punctiforme* is capable of forming vegetative cells, heterocysts, dormant cells that are called akinetes as well as motile hormogonia [19], while for *T. elongatus* the genetic prerequisites for cellular differentiation have not been found [45]. This implies that the Dps protein containing the His-type FOC might have a general role among those cyanobacteria that possess it. This hypothesis is also reinforced by the different ecological context in which *T. elongatus* and *N. punctiforme* live. While *T. elongatus* is a thermophile in hot springs with optimal growth temperatures at 55 ˚C [5], *N. punctiforme* was isolated from coralloid roots of an Australian cycad [19]. To reveal what specific function the His-type FOC has in *N. punctiforme*, we have collected data to characterize and also to compare it to the canonical Dps that were earlier characterized in *N. punctiforme*. Our working hypothesis is that the five NpDps complement each other in their roles to comply with the interdependent processes iron homeostasis and ROS protection.

## Iron binding in NpDps4 and its physiological role

By crystal structure analysis we identified two Fe atoms that bind to the FOC in NpDps4. This is in agreement with the general hypothesis that Dps proteins exhibit FOCs that coordinate two Fe atoms [46]. However, only a few Dps crystal structures have shown two Fe atoms bound to the FOC [29–31]. More often, Dps crystal structures have been reported to be occupied only at site A by an iron atom. It was argued that the weaker coordination affinity at site B would lead to an empty FOC site B or a coordination to a water molecule [31,43,47,48]. In our spectroscopic analysis we observed that NpDps4 catalyzed $Fe^{2+}$ by $O_2$, but the presence of NpDps4 did not alter the oxidation rate of $Fe^{2+}$ reacting with $H_2O_2$. This is different from the earlier studied canonical Dps from *N. punctiforme* (NpDps1-3) in which $Fe^{2+}$ is oxidized by $H_2O_2$ and not by $O_2$ (Howe et al 2018). The NpDps 1–3 dependent $H_2O_2$ oxidation was shown with the same spectroscopic experimental strategy as the one used for NpDps4 [49]. Typically Dps proteins reduce both $O_2$ and $H_2O_2$, with an up to 100-fold higher affinity towards $H_2O_2$ [4]. Our results do not exclude that NpDps4 might be able to use $H_2O_2$ under other conditions then the once investigated, but still imply that NpDps4 might not be involved in regulating the intracellular $H_2O_2$ level, but rather act as an $O_2$ scavenger. NpDps4 would therefore be able to regulate the $O_2$ concentration, and the intracellular $Fe^{2+}$ pool. For a heterocystous cyanobacterium like *N. punctiforme*, the regulation of both $Fe^{2+}$ and the $O_2$ might be vital under certain circumstances, especially in conditions that promote oxidative stress. We have earlier discovered that NpDps4 is more abundant in heterocysts than in vegetative cells [18], and now suggest that its ability to reduce $O_2$ via $Fe^{2+}$ might be crucial in heterocysts. Heterocysts are known to sustain a microoxic environment which is needed for the activity of the $O_2$-sensitive nitrogenase [19]. There are well characterized intracellular mechanisms that reduce the $O_2$ concentration in heterocysts such as the increased respiration rate, a non-$O_2$ producing photosystem II and multiple cell walls impeding $O_2$ diffusion from the outside [19]. However, it may be that NpDps4 acts as an additional valve to regulate the $O_2$ concentration if a sufficient $Fe^{2+}$ pool is available. Similar to NpDps4, the BaDps1 from the facultative aerobic *Bacillus anthracis*

was identified to only utilize $O_2$ for $Fe^{2+}$ oxidation. *B. anthracis* possesses another Dps, the BaDps2, which could effectively utilize both the oxidants $O_2$ and $H_2O_2$ [44]. The authors argued that the functional separation for the Dps may enable a better environmental adaption to a variety of $O_2$ concentrations. We have earlier studied the oxidant preferences of NpDps1-3 and found that all three utilize $H_2O_2$, but not solely $O_2$ for $Fe^{2+}$ oxidation [20]. Based on sequence comparisons, these three NpDps comprise a canonical FOC and it might be that amino acid exchanges in the coordination sphere of the FOC, e.g. His vs Asp, could lead to different oxidant preferences towards $H_2O_2$ or $O_2$ as suggested for the DpsA from *T. elongatus* [14]. This potential structure-function relationship could serve as a basis for future studies. For the studied NpDps it is clear that both the FOC coordination spheres and the respective physiological roles of NpDps1-3 clearly distinguishes them from NpDps4 [13,21]. The physiological significance of the differential oxidant preferences among the NpDps remains to be explored, but some structural features such as the pore structures might influence the oxidant preference. These pores guide $Fe^{2+}$, but also its potent oxidants $H_2O_2$ and $O_2$ into the internal protein cavity.

## The atypical ferritin-like and Dps-type pores in NpDps4

Dps proteins usually exhibit four ferritin-like and four Dps-type pores connecting the exterior with the inner protein cavity. The pores will be discussed individually starting with the ferritin-like pore. In the ferritin-like pore from NpDps4 two positively charged Arg are flanked by Glu (cavity, Glu152, Arg148, Arg145, Glu140 towards exterior). These two Arg could potentially electrostatically shield the pore from metal entrance. In canonical Dps the existence of positively charged amino acids across the channel has rarely been reported. Instead, several conserved negatively charged Asp and to a lesser extent from Glu have been studied [50]. In *L. innocua* Dps (LiDps) it has been shown that negatively charged amino acids are important for $Fe^{2+}$ translocation into the protein cavity as they attract the ions by charge [34,51]. Although poorly studied, there are other Dps proteins with a similar pore interior as identified for NpDps4. The Dps2 from *Mycobacterium smegmatis* (MsDps2) exhibits a similar amino acid channel composition. In MsDps2 a pair of His is flanked by two Asp (cavity, Asp138, His141, His126, Asp127, towards exterior). The authors suggested that the His, which are not conserved among Dps, could influence the conformations of the Asp thereby favoring iron uptake or its release out of the Dps. [52]. Also for the HsDpsA a similar amino acid arrangement (cavity, Glu154, Arg153, His150, Glu141, towards exterior) was found, although in a closed state, in which the translocation of $Fe^{2+}$ was unlikely [31]. However, structural information on Dps pores originates from static crystal structure analysis. It is therefore unclear whether molecular dynamics in the protein structure underline the pore opening and closing mechanisms.

The Dps-type pore of NpDps4 is lined with polar hydroxyl groups and negatively charged carboxylates from Ser61, Glu58 and Thr175. Also this finding deviates from the common picture of typical Dps proteins, since Dps-type pores are usually of hydrophobic character and the participating amino acids are not conserved. Typically, the Dps-type pore of canonical Dps has not been associated with metal ion translocation [2,5]. However, in atypical Dps such as the HsDpsA, similarities in the amino acid composition of the NpDps4 Dps-type pore can be found. HsDpsA has an even more negatively charged character (cavity, Glu56, Glu171, Asp172, towards exterior) as compared to NpDps4. In HsDpsA, Fe atoms were identified by crystal structure analysis to be bound within the Dps-type pore, indicating the Dps-type pore has a role in iron translocation [31]. Notably, the exterior of the Dps-type pores in NpDps4 is capped by three C-terminal residues, but leaving an opening to the water-loaded pore. Dps-type pore has been reported to be flexible in their aperture [5], but many have also been found in a closed state. Similar to the cap-like structure in NpDps4, C-terminal extensions were

found in the *Deinococcus radiodurans* Dps1 (DrDps1), although blocking the entry of the pore [53]. But more often the blockade is caused by bulky side chains within the channel impeding a possible metal ion translocation [5,16,52,54–56]. The interface at the Dps-like pore has additionally been identified to be a crucial structural element that evolutionarily links maxiferritins with Dps proteins [57]. It was shown that a single mutation at this interface could render the dodecameric structure of a Dps protein into a maxiferritin during protein crystallization.

Static analyses via crystal structural determination is accompanied with clear limitations on answering crucial question about dynamic processes in Dps proteins. In the future other techniques are required to verify previously suggested conformation-regulated opening and closing mechanisms of the two pore types. Furthermore, the structural variety among Dps-type and ferritin-like pores indicates the need for further structural classification. The pore structures in Dps proteins have been assigned as the gate keepers for metal ion translocation. But not only iron has been identified to travel into the protein cavity. $Zn^{2+}$ has often been found to bind to the FOCs.

## Zinc inhibits NpDps4 oxidizing $Fe^{2+}$ at the His-type FOC

One Zn atom was coordinated to the FOC site A in the $Zn^{2+}$-soaked crystals of NpDps4. The observation of Zn-binding at the FOC was not surprising, as other Dps proteins were also found to coordinate one Zn atom at site A [35] or two Zn atoms occupying both site A and site B [14,58] at the FOC. The FOC in Dps structures contains His, Asp and Glu, which are typical amino acids for Zn-binding sites in other enzymes that utilize zinc ions as a cofactor [59,60]. But is zinc really a cofactor in Dps proteins as suggested for the TeDpsA [14]? In our spectroscopic experiments we observed that $Zn^{2+}$ efficiently inhibited NpDps4 catalyzing the $O_2$ mediated $Fe^{2+}$ oxidation. This effect has also been seen for other Dps e.g. DrDps1 [37], *Listeria innocua* Dps (LiDps) [36] and *Streptococcus suis* Dps (SsDps) [35]. By contrast, the Zn-bound TeDpsA was observed to have nearly equal oxidant preferences towards $H_2O_2$ and $O_2$ for $Fe^{2+}$ oxidation catalysis. Notably, Dps usually oxidize $Fe^{2+}$ ~ 100 times faster with $H_2O_2$ as compared to with $O_2$ and have been associated to protect the organism against $H_2O_2$ stress [4]. The authors argued that the effect of $Zn^{2+}$ on the TeDpsA might be crucial for an effective regulation of the $O_2$ concentration in *T. elongatus* [14]. However, they could not obtain Zn-free TeDpsA protein material to characterize its activity serving as a control. The structural similarity between the FOC of TeDpsA and NpDps4 suggests similar functions, but the different biochemical role of $Zn^{2+}$ remains unclear.

Besides $Zn^{2+}$, Dps proteins appear to notoriously bind (at the FOC or at other binding sites) to a large variety of metal ions such as $Cd^{2+}$ [43], $Co^{2+}$ [50,53], $Mn^{2+}$ [50,61], $Ni^{2+}$ [50] and $Cu^{2+}$ [50]. Therefore it has been hypothesized that they may act as intracellular metal ion sponges [30]. High concentrations of these metals could lead to stress conditions causing damages to the organism as shown for *N. punctiforme* [62]. In this regard, ferritins have been suggested to act as zinc detoxification agents delivering a certain stress resistance [63]. For *S. suis* it has been hypothesized that its Dps may also deliver a protective mechanism against zinc at toxic concentrations, even if that would mean the inhibition of its FOC [35]. In *N. punctiforme* $Zn^{2+}$ concentrations above 18 μM have been shown to be cytotoxic [62]. It is not known whether NpDps4 is involved in $Zn^{2+}$ regulation in *N. punctiforme* [64]. It remains unknown whether Dps proteins have a native function in regulating other metals than $Fe^{2+}$, but it might be that their metal-binding properties is simply an encompassed effect of their innate design to attract and sequester iron.

We have collected data that resolves different biochemical aspects of the non-canonical NpDps4 suggesting it to be an $O_2$-scavenger in heterocysts of *N. punctiforme*. Through crystal

   

structure analysis and sequence alignment, we have classified a new type of FOC, the His-type FOC. It comprises a unique composition, within Dps structures, that can be broadly found across the cyanobacterial phylum. Whether the His-type FOC is a feature that is confined to cyanobacterial Dps and connected to the oxygenic photosynthesis of cyanobacteria needs to be further investigated.

## Experimental procedures

### Cloning, overexpression and protein purification

The *Npdps4* gene encoding residues 1–184 of the protein was PCR amplified from genomic *N. punctiforme* ATCC 29133 by using primers Forward_NpDps4 (*NcoI* site in bold); 5′– TTTTT**CCATGG**CTGAAACGCAAA–3, Reverse_NpDps4 (*Acc65I* site in bold); 5′AAAAA**GG TACC**CTAGCTTTGAGCCGCTTG3′. The PCR product was digested with *Acc65I* and *NcoI* and ligated into the equivalent sites of the pET-His1a expression vector (kindly provided by G. Stier, EMBL, Germany). The final construct encodes MKHHHHHHP-Dps$_{1-184}$. For cloning reasons residue number 2 was mutated from Ser to Ala. The plasmids were transformed into *E. coli* DH5-α and subsequent colonies selected on kanamycin plates. The positive clones were verified by DNA sequencing by using primer Seq-NpDps4; 5'GGTAATGTATATGACAAT CCCGTGTTG3'. The protein was overexpressed in *E. coli* BL21 (DE3) at 37 ˚C in Luria Broth supplemented with 50 μg mL$^{-1}$ kanamycin. When the cultures reached an OD$_{600}$ of 0.4, the temperature was lowered to 28 ˚C and the expression was induced with 0.4 mM IPTG after which the cultures were grown for additional 5 h. Cells were harvested by centrifugation at 5300 x g and the pellets were frozen at -80 ˚C. Cell pellets were suspended in 50 mM NaH$_2$PO$_4$ pH 8.0, 300 mM NaCl and 10 mM imidazole supplemented with EDTA-free protease inhibitor cocktail (Roche) and 0.5% triton X-100. The suspension was lysed on ice by sonication and cellular debris was removed by centrifugation at 39000 x g for 60 min. The supernatant was loaded onto a column packed with Ni-NTA agarose (Qiagen). The protein was washed in 50 mM NaH$_2$PO$_4$ pH 8.0, 300 mM NaCl and 20 mM imidazole and eluted with 50 mM NaH$_2$PO$_4$ pH 8.0, 300 mM NaCl and 300 mM imidazole. The buffer was exchanged to 20 mM Tris-HCl pH 8.0, 200 mM NaCl, and 0.5 mM EDTA. The protein was further purified by size-exclusion chromatography using a HiLoad 16/60 Superdex 200 prep-grade column (Amersham Biosciences) in the same buffer. The protein purity was judged by SDS-PAGE and concentrated to 30 mg mL$^{-1}$ in 20 mM Tris-HCl pH 8.0, using an Amicon Ultra centrifugal filter device (Millipore).

### Crystallization and data collection

Initial crystallization trials were performed by the sitting-drop vapour-diffusion method in a 96-well MRC-crystallization plate (Molecular Dimensions) using a Mosquito (TTP Labtech) pipetting robot. Droplets of 0.1 μL protein solution at 10 mg mL$^{-1}$ were mixed with an equal volume of reservoir solution using screens from Molecular Dimensions (PACT, PGA, Structure Screen I/II and MIDAS) at room temperature. Crystals appeared in several conditions. The final crystallization condition was optimized to 25% SOKALAN HP 66, 0.1 M HEPES pH 7.0 and 0.2 M NaOAc. Rod shaped crystals grew in a few days. The crystals were soaked for a few seconds in mother liquor solution supplemented with 20%-glycerol before they were flash cooled in liquid N$_2$ and stored until data collection. For obtaining metal complexes the crystals were transferred to a cryo solution containing 15%-glycerol and 10 mM of the metal ion solution (FeSO$_4$ and ZnSO$_4$ respectively). In this solution the chelating SOKALAN HP 66 was exchanged for 25%-PEG4000. Diffraction data of the metal-free and metal-treated proteins were collected at beamlines ID29 and ID23-1 respectively at the European Synchrotron

Radiation Facility, Grenoble, France using Pilatus 6M-F detectors. Diffraction images were processed with XDS and scaled with AIMLESS from the CCP4 program suit [65]. Relevant processing statistics are summarized in Table 1.

## Structure determination

The metal-free structure was determined using the molecular replacement option in auto rickshaw [66] using the Dps structure from *T. elongatus* (PDB ID: 2VXX) [14]. Density modification and automatic model building were performed using AutoRickshaw and ARP/wARP [67]. For refinement, 5% of the reflections were removed for the calculation of $R_{free}$. The model was further built using rounds of manual building in COOT [68] and refinement using phenix.refine [69]. Four molecules were found in the asymmetric unit, which represents a third of the biological unit, the dodecameric hollow sphere. The metal bound structures were solved by phaser [70] using the metal-free structure as the start model. In the last rounds of refinement translational-libration-screw (TLS) [71] refinement was used, treating each

**Table 1. Data collection and refinement statistics.**

| Data collection | Metal-free NpDps4 | NpDps4 Fe$^{2+}$ | NpDps4 Zn$^{2+}$ |
|---|---|---|---|
| Space group | P6$_3$ | P6$_3$ | P6$_3$ |
| Cell dimensions (Å) | 101.9 101.9 146.2 | 100.3 100.3 145.6 | 100.9 100.9 144.3 |
| Wavelength (Å) | 0.984 | 0.972 | 0.972 |
| Resolution (Å)* | 48.75–1.59 (1.64–1.59) | 48.55–1.88 (1.95–1.88) | 48.09–2.39 (2.48–2.39) |
| Total reflections* | 1174206 (113526) | 288333 (28591) | 277574 (26712) |
| Unique reflections* | 115112 (11130) | 67029 (6512) | 32761 (3151) |
| I/σ(I)* | 19.6 (3.8) | 12.3 (1.7) | 13.9 (1.2) |
| R$_{merge}$* | 0.064 (0.482) | 0.066 (0.993) | 0.115 (1.579) |
| R$_{pim}$* | 0.022 (0.165) | 0.035 (0.529) | 0.063 (0.853) |
| Completeness (%)* | 99.7 (98.5) | 99.6 (99.0) | 99.8 (98.5) |
| Multiplicity* | 10.2 (10.2) | 4.3 (4.4) | 8.5 (8.5) |
| CC(1/2)* | 0.999 (0.921) | 0.997 (0.673) | 0.997 (0.540) |
| **Refinement** | | | |
| No reflections in working set (test set) | 110427 (4654) | 65014 (2005) | 31108 (1622) |
| R$_{work}$ (%) | 13.01(17.34) | 15.76 (28.39) | 16.87(28.69) |
| R$_{free}$ (%) | 14.62 (21.44) | 19.31(36.86) | 21.41(32.74) |
| Average B-factors (Å$^2$) | | | |
| Protein | 22.7 | 40.1 | 57.5 |
| Water | 32.3 | 46.4 | 51.2 |
| Metal | N/A | 55.0 | 77.6 |
| Ligands | N/A | 64.3 | 70.0 |
| RMSD from ideal | | | |
| Bond lengths | 0.009 | 0.007 | 0.008 |
| Bond angles | 1.170 | 0.955 | 1.084 |
| **Ramachandran plot** | | | |
| Most favoured (%) | 97.4 | 97.9 | 97.0 |
| Outliers (%) | 0.4 | 0.0 | 3.1 |
| PDB code | 5HJF | 5HJH | 5I4J |

*Values within parentheses are for the highest resolution shell.

molecule as an individual TLS group. The quality of the model was analyzed with MolProbity in PHENIX [72]. Crystallographic statistics are summarized in Table 1. Figures were drawn with CCP4MG [73]. The X-ray coordinates and structure factors have been deposited in the Protein Data Bank under accession codes 5HJF, 5HJH and 5I4J.

## Spectroscopic analyses of NpDps4

Spectroscopic experiments of $Fe^{2+}$ oxidation by $O_2/H_2O_2$ were performed on a Cary 5000 spectrophotometer (Varian). Samples were measured in 1 cm standard quartz cuvettes. All experiments were done at room temperature. The time-dependent absorbance was recorded at 310 nm, which corresponds to $Fe^{3+}$ formation [4,34]. During all experiments, reaction solutions were maintained aerobic. The reaction buffer contained in all reactions 5 mM succinate and 50 mM NaCl, at pH 6.0. Where indicated, 0.5 μM (dodecamer concentration) of purified NpDps4 protein was added. As the protein addition caused a slight background absorbance increase, the absorbance was set to zero and no further change during prolonged protein incubation was detected.

When the reactivity between $Fe^{2+}$ and $O_2$ (reaction mixture was exposed to air) were investigated, freshly prepared and $N_2$-sparged 24 μM $FeSO_4$ was added to the solution and the absorbance at 310 nm was monitored between 10 min and 8 hours depending on the experimental scheme. To investigate the reactivity of $Fe^{2+}$ and $H_2O_2$, 16 μM $H_2O_2$ was incubated with NpDps4 between 2–10 min. Subsequently 24 μM $Fe^{2+}$ was added and the oxidation was followed as absorbance at 310 nm.

To study the effect of $Zn^{2+}$ on the oxidation of $Fe^{2+}$ by $O_2$ (reaction mixture was exposed to air) 12 μM $ZnSO_4$ was added to the reaction mixture in two separate experiments. In the 1st, $Zn^{2+}$ was incubated with the protein material for 8 min and then $Fe^{2+}$ was added to the reaction mixture. In a 2nd experiment $Fe^{2+}$ was first added to the protein and after 5 min $Zn^{2+}$ was added. The absorbance was followed at 310 nm.

After the addition of any chemical the reaction volume was quickly mixed by multiple resuspending. Controls of identical experiments without the addition of protein were performed. Absorption spectra (200–800 nm) were frequently recorded to check whether protein solubility was affected. The absorbance data presented in this study were reconstructed in their time-axis as absorbance spectra were recorded in between subsequent absorbance traces over time. The data were analyzed with OriginPro 2016G software (OriginLab, Northampton, MA). All data presented originate from two replicate experiments (S1 File).

## Sequence alignment

All cyanobacterial amino acid sequences belonging to the Dps protein family were downloaded in November/December 2018 from the Cyanobase database [24] (http://genome.microbedb.jp/cyanobase/). To find the homologous Dps sequences, the in-built alignment function within Cyanobase database was utilized. In addition to the cyanobacterial sequences, a selection of non-cyanobacterial sequences (bacterial and archaeal) belonging to the Dps protein family were downloaded from the KEGG database (https://www.genome.jp/kegg/). These sequences were aligned with Clustal Omega software [25] (https://www.ebi.ac.uk/Tools/msa/clustalo/) using default settings. Cyanobacterial sequences were chosen to represent members of recent phylogenetic classification among cyanobacteria [41]. Information about the morphology (unicellular or filamentous), the capacity of forming heterocysts for $N_2$ fixation and the origin of isolation (freshwater or marine) of the cyanobacterial species were included [41].

### Protein Data Bank accession codes

The refined structures have been deposited in the RCSB Protein Data Bank and are available under accession codes 5HJF for the metal-free, 5HJH and 5I4J for the iron and zinc soaked crystal structures, respectively (S2–S4 Files).

## Supporting information

**S1 Fig. Interactions at the BC helices.** Hydrophobic interaction between the BC helix from two subunits depicted in grey and purple, respectively. Residues Phe95, Leu98 and Ala99 are depicted as stick models.
(TIFF)

**S2 Fig. Anomalous density at the FOC.** An anomalous difference map (3σ) in green indicates the position of the metals. A. The anomalous density confirms two metals bound (A- and B-site) in the Fe-soaked protein. B. In the Zn-soaked protein the anomalous density indicate one metal with high occupancy bound in the A-site. The density also suggests the possibility of a Zn-atom in the B-site, however this metal has not been modelled due to low occupancy.
(TIF)

**S3 Fig. Argon experiment.** The reaction mixture containing 0.5 μM NpDps4, 50 mM NaCl in 5 mM succinate at pH 6.0 was continuously aerated with Argon gas. $Fe^{2+}$ addition (24 μM) performed under Argon atmosphere. Fe addition and aeration with air indicated with arrows.
(TIF)

**S1 Table. Coordination distances table.** The coordination distances of the A and B sites.
(DOCX)

**S1 File. Raw data from spectroscopic analyses of NpDps4.**
(ODS)

**S2 File. 5hjf_full_validation.** Full wwPDB X-ray structure validation report for 5hjf.
(PDF)

**S3 File. 5hjh_full_validation.** Full wwPDB X-ray structure validation report for 5hjh.
(PDF)

**S4 File. 5i4j_full_validation.** Full wwPDB X-ray structure validation report for 5i4j.
(PDF)

## Acknowledgments

The authors thank the staff at the European Synchrotron Radiation Facility, Grenoble, France, beamlines ID23-1 and ID29 for assistance with data collection. We are grateful to Dr. Burkhard Zietz (Uppsala University, Sweden) for his valuable contribution.

## Author Contributions

**Conceptualization:** Karin Stensjö.

**Data curation:** Christoph Howe, Karina Persson.

**Formal analysis:** Christoph Howe, Felix M. Ho, Karina Persson.

**Funding acquisition:** Karin Stensjö.

**Investigation:** Christoph Howe, Vamsi K. Moparthi, Karina Persson.

**Methodology:** Christoph Howe, Vamsi K. Moparthi, Felix M. Ho, Karina Persson.

**Project administration:** Karin Stensjö.

**Resources:** Karin Stensjö.

**Supervision:** Felix M. Ho, Karin Stensjö.

**Validation:** Christoph Howe, Karin Stensjö.

**Visualization:** Christoph Howe, Karina Persson.

**Writing – original draft:** Christoph Howe.

**Writing – review & editing:** Vamsi K. Moparthi, Felix M. Ho, Karina Persson, Karin Stensjö.

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
