## [Decision Letter · Decision Letter 0]

26 Jun 2019

PONE-D-19-15019

The Dps4 from Nostoc punctiforme ATCC 29133 is a member of His-type FOC containing Dps protein class that can be broadly found among cyanobacteria

PLOS ONE

Dear Dr. Stensjö,

Thank you for submitting your manuscript to PLOS ONE. After careful consideration, we feel that it has merit but does not fully meet PLOS ONE’s publication criteria as it currently stands. Therefore, we invite you to submit a revised version of the manuscript that addresses the points raised during the review process.

Both reviewers raise a number of points regarding the interpretation of the kinetic and structural data, which need to be fully addressed. If you feel that you can change the manuscript in order to address these points, we would be happy to receive a revised version.

We would appreciate receiving your revised manuscript by Aug 10 2019 11:59PM. To enhance the reproducibility of your results, we recommend that if applicable you deposit your laboratory protocols in protocols.io, where a protocol can be assigned its own identifier (DOI) such that it can be cited independently in the future. For instructions see: http://journals.plos.org/plosone/s/submission-guidelines#loc-laboratory-protocols

We look forward to receiving your revised manuscript.

Kind regards,

Inês A. Cardoso Pereira, Ph.D.

Academic Editor

PLOS ONE

Journal Requirements:

Reviewers' comments:

Reviewer's Responses to Questions

**Comments to the Author**

1. Is the manuscript technically sound, and do the data support the conclusions?

Reviewer #1: Partly

Reviewer #2: Yes

2. Has the statistical analysis been performed appropriately and rigorously? 

Reviewer #1: No

Reviewer #2: N/A

3. Have the authors made all data underlying the findings in their manuscript fully available?

Reviewer #1: Yes

Reviewer #2: Yes

4. Is the manuscript presented in an intelligible fashion and written in standard English?

Reviewer #1: Yes

Reviewer #2: Yes

5. Review Comments to the Author

Reviewer #1: The current manuscript focused on the structural and kinetic studies of Dps4 from a cyanobacteria. Interestingly Nostoc punctiforme, the phototrophic and filamentous cyanobacterium encodes for five different Dps, but the specific function of each Dps it is not fully yet stablished. Thus, the work present in the current manuscript, contribute to elucidate the function of the NpDps4. Nevertheless, there are a several modifications that have to be included in the manuscript, mainly regarding the kinetic analysis of the data, which was not properly performed; and a more detailed crystal structure analysis.

The authors should be more carefully with the scientific language, please rephrase the following sentences.

Page 6, line 147 - “meet thus creating a pore.” - The pore is not created, it is formed, and word “meet” should be avoided. Please rephrase.

Page 7, line 158 - “of three monomers meet”. Again the word “meet” should be avoided.

Comments:

Page 5-9, line 123-204 –Although the overall fold is the same in the three structures presented (5HJF, 5HJH, 5I4J), the authors should present the rmsd between the different structures.

Besides the difference in the ferroxidase center that the authors comment, is there any structural difference in the crystal structures induced by the presence of the different metals, for instance in the 3-fold pores, or in other part of the structure.

Page 6 and 7, line 147-153- The authors described the ferritin-like pore of the NpDps4, but they do not mentioned if there is any water or metal located in this pore.

Page 7, line 158-164 – The authors describe the Dps-like pore in the NpDps4, and they mention that it contains a “small cap-like structure on the outside of the protein”. However it is not clear if the pore is blocked by any residue from the C-terminal tails that are at the top of the pore, and if the nature of these residues is negative or positive. A more detailed description should be included.

Page 8, line 188, 195- The authors mentioned that the distance between Fe in site A and Fe in site B is 3.5 Å, and that there is one water molecule coordinating both irons. Is this a bridging molecule, if so it is not clear how can be at a distance of 2.2 and 2.3 Å from site A and B respectively, since the distance between the two irons is 3.5 Å. This part should be clarified.

Page 7-9, lines 173-205 – In order to understand if the Fe in site B is more loosely coordinated, and if the distances between the metals and the coordinating residues are similar in the structure with Fe or Zn, a Table with the coordination distances between the different coordinating residues and the metal ions for the different structures should be included. The authors should compare the distances between the coordinating residues and Fe in site A, and those for Fe in site B.

Page 8, lines 198-204; Page 10, lines 252-269 – The authors have determined the crystal structure of NpDps4 in which the crystals were independently soaked with a solution of FeSO4 or ZnSO4. The structures showed that the iron was bound in two position, site A and B, while in the case of the structure with zinc, the metal was bound to a similar structural position as the Fe in site A. Moreover, by kinetic studies the authors showed that the zinc inhibits the iron oxidation rates. In order to understand if the “di-iron” center formed upon iron addition is not maintained in the presence of zinc, or if it is form a mixed di-center with Zn and Fe, it would be interesting to determine the crystal structure, in which the crystals were first soaked with an iron solution followed by a zinc solution.

Page 10, 226-240 – Iron oxidation assays. The authors mentioned the increase of absorbance at 310 nm which is related with the oxidation of iron by the protein. Nevertheless, the authors should present a proper analysis of the data, the initial rate should be determined for each assay, with the corresponding error bar.

Moreover, the authors only present one condition tested, in which they have used 0.5 uM with 24uM Fe, corresponding to a ratio of 48 Fe/dodecamer and 4Fe/monomer. A proper kinetic study should be included, using different Fe/dodecamer from 12 to the maximum capacity per dodecamer. In addition, for each assay the iron oxidation rates have to be determined, which is not presented in the current version of the manuscript. As an example “Abs310nm 0.014 after 2 min” (page 10, line 230) is not a rate.

Page 10, lines 241-251 – The authors also address the iron oxidation by using hydrogen peroxide. Although they do not see any difference with and without the protein, in order to account only for the contribution of hydrogen peroxide, have the authors performed the assays for iron oxidation by hydrogen peroxide under anaerobic conditions. As mentioned before, the data should be presented with the error bars, and they have to present iron oxidation rates.

Page 11, lines 258-260; Fig 6A- From Fig 6A and from the corresponding legend it is clear that the protein is able to oxidize the iron (black line) but if zinc was added before, the iron oxidation is inhibited. Nevertheless, the corresponding phrase, in the text (page 11, lines 258-260) is not clear, please rephrase it.

Page 15-17, lines 368-415- The authors present an analysis of different amino acid sequences of Dps, and they proposed a His-type FOC. The others NpDps instead of His63 present an aspartate residues, moreover as it states in the Introduction part (page 4, lines 91-92) the NpDps1-3 catalyzes the iron oxidation with hydrogen peroxide and not with oxygen, while NpDps4 is the other way around. So, it would be interesting if the authors could comment on the oxidation rates for the other NpDps or even for “canonical” Dps that contain a aspartate residue and their specificity of hydrogen peroxide vs oxygen and histidine vs aspartate.

Page 19, line 481 – Please define DrDps1, since it is the first time it is mentioned in the text.

Page 20, lines 514-515 – The authors have wrote that “ferritins have be found to act as zinc detoxification agents delivering a certain stress resistance”- do the authors have any evidence that the NpDps4 stores Zn or was only tested for the inhibition of the iron oxidation.

Figure 2 – The different subunits should be in different colours, similar to the one presented in Figure 4. In addition, the corresponding residues should use the same colour as the subunit, maintaining the red for the oxygen and blue for nitrogen.

Figure 3 – For clarity, figure 3A should be zoom-in and the residues should be in the same colour as the corresponding subunit, maintaining the red for the oxygen and blue for nitrogen.

Figure 7 – From the figure it was not clear the representation of “lengths of α-helices refer to the crystal structure of NpDps4 and are indicated by black bars and red letters.” In the figure legend the explanation for the different His-type FOC should be induced, namely the definition of Nx, SynPro, LPP-B, AcTh, Osc, SPM.

Figure 7 – Since N. punctiforme contains five different Dps, is there any reason why NpDps5 was not included in the Multiple sequence alignment presented in Fig. 7. Moreover, for clarity for the reader, the figure should include the name of the organism instead of the protein code. The figure legend should have the information for each protein code and the corresponding name of the organism, as it is currently.

Table 1 – The authors should comment on why they did not process the data of Metal-free NpDps4 structure to a higher resolution based on their values of I/σ(I).

To be consistent the authors should present the values for highest resolution shell for the Rwork and Rfree.

Reviewer #2: The authors Howe et al., in their work about the fourth Dps from Nostoc punctiforme has tried to unravel its role in the organism and propose a new class of Dps, having His-type Ferroxidation centres unique to cyanobacteria. They also claim the NpDps4 uses O2 over H2O2 to oxidise iron. Also, crystal structures of apo, iron bound and zinc bound Dps has been analysed and they conclude by spectroscopic assays that binding of Zinc to the FOC inhibits ferroxidation.

While it’s interesting to note the presence of a new type of ferroxidation site in cyanobacterial Dps, the authors could not make convincing theories about its possible biological significance. There are also some major points that the authors need to address regarding their other findings to make their claims credible.

The following points are raised in the order they occur in the text:

Line 142

The residues Phe95, Leu98 and Ala99 are mentioned in the text but not marked in the figure.

Line 157

… acid of the trio …

Lines 168-169

“each symmetrical amino acid of the trio is indicated only once”, but Figure 3B shows amino acids from all three monomers related by three-fold.

Line 174

How does the authors know SOKALAN66 is a metal chelating agent?

Line 176

Is there any experimental evidence whether it is really Zinc or Iron bound in the crystal structure like anomalous X-ray scattering or even a simple detection of Zinc ions on a gel?

Line 177

It would be more desirable to show electron density maps to show the fitting of ion atoms in Figure 4

Line 189

Since the FOC is formed not only of residues from two monomers related by two-fold, but also from a third monomer it is also unusual for Dps proteins.

Lines 213-214

The sentence can be phrased better

Line 220

What is the rationale of using 24 mM Fe2+ and 0.5 mM NpDps?

Lines 241-250

If the ferroxidation experiments using H2O2 were done under aerobic conditions, how does the authors discount for the effect of O2 on oxidation and especially as they’re comparing the ferroxidation properties between O2 and H2O2. It is strange that the graph doesn’t show any increase in ferroxidation under aerobic conditions, as in the above paragraph they show that oxygen can be used for ferroxidation by NpDps4.

Lines 262-263

Does the inhibition of ferroxidation by Zinc related to it causing protein aggregation as it is mentioned in line 263 that Zinc negatively impacts protein solubility?

Lines 282-288

Addition of Zinc indicated by arrow is misleading in Figure 6B as the black graph shows the reaction without Zn addition. In line 287 the authors mean figure 6B and not 6A?

Line 302

Figure 7 is illegible, and nothing can be inferred from it. So, the authors most important claim that the his-type FOC is conserved across cyanobacteria cannot be ascertained at all from the figure.

Line 392

It’s not clear to the reader what is ‘the specific function’ in cyanobacteria.

Line 470

Recent developments on the dps-like interface and its role as an evolutionary switch between dps and ferritins has to be mentioned when discussing the role of this interface.

Lines 620-621

During the ferroxidation experiment the authors talk of mixing the reaction by resuspension. How do they account for the reaction time that has elapsed during this process? Wherever possible cuvettes with stirrers should be used as these reaction rates can be pretty fast, especially the rates with H2O2.

Line 626

Ideally kinetic data should be evaluated from 3-4 experiments and at least two different protein preparations.

6. PLOS authors have the option to publish the peer review history of their article (what does this mean?). If published, this will include your full peer review and any attached files.

Reviewer #1: No

Reviewer #2: No

---

## [Author Response · Author response to Decision Letter 0]

16 Jul 2019

Uppsala, July 16, 2019

Dear Editor, Dr Pereira,

Enclosed, please find a revised version of our manuscript entitled “The Dps4 from Nostoc punctiforme ATCC 29133 is a member of His-type FOC containing Dps class that can be broadly found among cyanobacteria”

By: Christoph Howe, Vamsi K. Moparthi, Felix M. Ho, Karina Persson and myself, which I hope now is acceptable for publication in PLOS ONE.

In this revised version we addressed all points raised by the Reviewers and Editor. As a result, the manuscript underwent significant changes and additions in both text and figures. We have tried our best to answer all the comments and to improve our manuscript accordingly. In particular, we modified some of the main figures and added requested information concerning a more detailed crystal structure analysis in Supplementary material. We also added missing experimental information and data to clarify details concerning our spectroscopic analysis. We believe that the revised data and results part are more clear and fully support the discussion and conclusions in the manuscript.

Our point to point response to the comments from both Reviewers can be found on the following pages. We appreciate very much time and effort that Reviewer(s) have spent and put in reading and commenting on our manuscript. We thank the reviewer(s) for their valuable suggestions which has improved the manuscript.

We have uploaded an annotated manuscript (word file) including track-changed text to allow for our changes to be easily found.

Yours Sincerely,

Karin Stensjö (karin.stensjo@kemi.uu.se), on behalf of all authors

 

Answers to Review Comments to the Author

Reviewer #1: The current manuscript focused on the structural and kinetic studies of Dps4 from a cyanobacteria. Interestingly Nostoc punctiforme, the phototrophic and filamentous cyanobacterium encodes for five different Dps, but the specific function of each Dps it is not fully yet stablished. Thus, the work present in the current manuscript, contribute to elucidate the function of the NpDps4. Nevertheless, there are a several modifications that have to be included in the manuscript, mainly regarding the kinetic analysis of the data, which was not properly performed; and a more detailed crystal structure analysis.

The authors should be more carefully with the scientific language, please rephrase the following sentences.

Page 6, line 147 - “meet thus creating a pore.” - The pore is not created, it is formed, and word “meet” should be avoided. Please rephrase.

-We have rephrased the sentence.

Page 7, line 158 - “of three monomers meet”. Again the word “meet” should be avoided.

-We have rephrased the sentence.

Page 5-9, line 123-204 –Although the overall fold is the same in the three structures presented (5HJF, 5HJH, 5I4J), the authors should present the rmsd between the different structures.

Besides the difference in the ferroxidase center that the authors comment, is there any structural difference in the crystal structures induced by the presence of the different metals, for instance in the 3-fold pores, or in other part of the structure.

-The rmsd are now included in the manuscript. We additionally contemplate on the differences of the three crystal structures in more detail.

Page 6 and 7, line 147-153- The authors described the ferritin-like pore of the NpDps4, but they do not mentioned if there is any water or metal located in this pore.

-In the metal-free and iron-bound enzyme there is a water molecule (low occupancy) located in the pore, equally distanced from Glu140 and Arg148. In the Zn-structure there are no ordered waters or ions in connection to the pore. 

Page 7, line 158-164 – The authors describe the Dps-like pore in the NpDps4, and they mention that it contains a “small cap-like structure on the outside of the protein”. However it is not clear if the pore is blocked by any residue from the C-terminal tails that are at the top of the pore, and if the nature of these residues is negative or positive. A more detailed description should be included.

-The pore is not blocked by the C-terminal residues. The sequence of the five final residues in the models is FVQAA. The five final residues that form the cap leave an opening to the pore, e.g. the pore is not blocked. Corresponding changes have been made in the text.

Page 8, line 188, 195- The authors mentioned that the distance between Fe in site A and Fe in site B is 3.5 Å, and that there is one water molecule coordinating both irons. Is this a bridging molecule, if so it is not clear how can be at a distance of 2.2 and 2.3 Å from site A and B respectively, since the distance between the two irons is 3.5 Å. This part should be clarified.

-The water molecule is not located on a straight line between the Fe-ions. The angle between the two Fe-ions and the bridging water molecule is 42° and 53°, respectively. And the the Fe – water – Fe angle is 95° This information was added to the manuscript.

Page 7-9, lines 173-205 – In order to understand if the Fe in site B is more loosely coordinated, and if the distances between the metals and the coordinating residues are similar in the structure with Fe or Zn, a Table with the coordination distances between the different coordinating residues and the metal ions for the different structures should be included. The authors should compare the distances between the coordinating residues and Fe in site A, and those for Fe in site B.

-In general the coordination distances are shorter in the A-site compared to the B-site. There are also differences in the different FOCs, especially regarding Glu82 which binds tightly to the metal in some FOCs whereas it has poorer density in some FOCs which also indicate a possible bidentate coordination. All the coordination distances are now presented in S1 Table.

Page 8, lines 198-204; Page 10, lines 252-269 – The authors have determined the crystal structure of NpDps4 in which the crystals were independently soaked with a solution of FeSO4 or ZnSO4. The structures showed that the iron was bound in two position, site A and B, while in the case of the structure with zinc, the metal was bound to a similar structural position as the Fe in site A. Moreover, by kinetic studies the authors showed that the zinc inhibits the iron oxidation rates. In order to understand if the “di-iron” center formed upon iron addition is not maintained in the presence of zinc, or if it is form a mixed di-center with Zn and Fe, it would be interesting to determine the crystal structure, in which the crystals were first soaked with an iron solution followed by a zinc solution.

-We agree with the referee that this would be an interesting study, and the results we present here regarding Zn inhibition does open up for such investigations. However, due to the likely degradation in crystal quality associated with changes to the soaking solutions, this should be carried out as a focused, separate crystallographic study. This is outside the scope of the present investigation.

Page 10, 226-240 – Iron oxidation assays. The authors mentioned the increase of absorbance at 310 nm which is related with the oxidation of iron by the protein. Nevertheless, the authors should present a proper analysis of the data, the initial rate should be determined for each assay, with the corresponding error bar. Moreover, the authors only present one condition tested, in which they have used 0.5 uM with 24uM Fe, corresponding to a ratio of 48 Fe/dodecamer and 4Fe/monomer. A proper kinetic study should be included, using different Fe/dodecamer from 12 to the maximum capacity per dodecamer. In addition, for each assay the iron oxidation rates have to be determined, which is not presented in the current version of the manuscript. As an example “Abs310nm 0.014 after 2 min” (page 10, line 230) is not a rate.

-The main purpose of the kinetics data here was to compare the Fe-oxidation behaviour of NpDps4 under different conditions. Precise quantitative kinetic analyses with iron oxidation rates were not the objective for this study. We believe that our data show the clear qualitative differences between the different samples and thus support our discussion and conclusions. We have thoroughly read through our text to ensure that we don´t claim to have determined any specific kinetic and inhibition parameters, which was outside the scope of our study. 

All data presented originate from the same protein material as was used for the crystallographic work. Two replicate experiments were performed for each analyses. All data from this experiments are now presented in S1 File.

Page 10, lines 241-251 – The authors also address the iron oxidation by using hydrogen peroxide. Although they do not see any difference with and without the protein, in order to account only for the contribution of hydrogen peroxide, have the authors performed the assays for iron oxidation by hydrogen peroxide under anaerobic conditions. As mentioned before, the data should be presented with the error bars, and they have to present iron oxidation rates.

-All our experiments have been conducted under aerobic conditions. Except for the results presented in supplementary material S3 Fig none of the experiments were performed under anaerobic condition. In regards to the Figure 5B showing the oxidation kinetics of the Fe2+ oxidation reaction with H2O2 under aerobic conditions in the presence of NpDps4, we cannot conclude that the observations are only from H2O2. It is likely that O2-mediated Fe2+ oxidation occurs in the presence of NpDps4, however this was not visible in the experiment. By using the same experimental strategy we previously observed a clear difference in the reaction between Fe2+ and H2O2 in the presence of NpDps1, NpDps2 and NpDps3 as compared to without protein (Howe et al. 2018). Thus the Fe2+ oxidation in NpDps4 is clarly different as compared to canonical Dps proteins.

Howe, C., Ho, F.M., Nenninger, A., Raleiras, P., Stensjö, K. (2018) Differential biochemical properties of three canonical Dps proteins from the cyanobacterium Nostoc punctiforme suggest distinct cellular functions. J. Biol. Chem. 293(43) 16635–16646.

Page 11, lines 258-260; Fig 6A- From Fig 6A and from the corresponding legend it is clear that the protein is able to oxidize the iron (black line) but if zinc was added before, the iron oxidation is inhibited. Nevertheless, the corresponding phrase, in the text (page 11, lines 258-260) is not clear, please rephrase it.

-We agree with reviewerthat the phrase is misleading and have now rewritten the whole paragraph for better clarity.

Page 15-17, lines 368-415- The authors present an analysis of different amino acid sequences of Dps, and they proposed a His-type FOC. The others NpDps instead of His63 present an aspartate residues, moreover as it states in the Introduction part (page 4, lines 91-92) the NpDps1-3 catalyzes the iron oxidation with hydrogen peroxide and not with oxygen, while NpDps4 is the other way around. So, it would be interesting if the authors could comment on the oxidation rates for the other NpDps or even for “canonical” Dps that contain a aspartate residue and their specificity of hydrogen peroxide vs oxygen and histidine vs aspartate.

-We agree with the reviewer that this is an interesting topic and have now included a short discussion about the putative importance of the exchange of amino acids. 

Page 19, line 481 – Please define DrDps1, since it is the first time it is mentioned in the text.

-We added the definition of DrDps 1 in the text.

Page 20, lines 514-515 – The authors have wrote that “ferritins have be found to act as zinc detoxification agents delivering a certain stress resistance”- do the authors have any evidence that the NpDps4 stores Zn or was only tested for the inhibition of the iron oxidation.

-We do not have any evidence that NpDps4 stores Zn2+. We have now changed the wording “have been found” to “have been suggested” as the referred literature did not include strong evidence for ferritins to be a zinc detoxification agent. 

Figure 2 – The different subunits should be in different colours, similar to the one presented in Figure 4. In addition, the corresponding residues should use the same colour as the subunit, maintaining the red for the oxygen and blue for nitrogen.

-Yes we agree and have made new figures accordingly. 

Figure 3 – For clarity, figure 3A should be zoom-in and the residues should be in the same colour as the corresponding subunit, maintaining the red for the oxygen and blue for nitrogen.

-Yes we agree and the figure has been revised.

Figure 7 – From the figure it was not clear the representation of “lengths of α-helices refer to the crystal structure of NpDps4 and are indicated by black bars and red letters.” In the figure legend the explanation for the different His-type FOC should be induced, namely the definition of Nx, SynPro, LPP-B, AcTh, Osc, SPM.

-We implemented the missing black bar and red letter A,B,BC,C and D in Figure 7 that represent the α -helices of NpDps4 and the helix annotation, respectively. For clarity, the sentence was also revised. The abbreviations for the different cyanobacterial classes have been added to the figure legend.

Figure 7 – Since N. punctiforme contains five different Dps, is there any reason why NpDps5 was not included in the Multiple sequence alignment presented in Fig. 7. Moreover, for clarity for the reader, the figure should include the name of the organism instead of the protein code. The figure legend should have the information for each protein code and the corresponding name of the organism, as it is currently.

-NpDps5 was not included into the sequence alignment as it was previously identified to comprise a Bfr-like sequence [Ekman et al. 2014]. We have identified similarities with the Dps-like structures from the archaea Pyrochoccocus furiosus [Ramsay et al. 2006] and Sulfolobus Solfataricus [Gauss et al. 2006]. NpDps5 requires a separate analysis. We have now included this explanation in the manuscript.

[Ekman et al.] Ekman M, Sandh G, Nenninger A, Oliveira P, Stensjö K. Cellular and functional specificity among ferritin-like proteins in the multicellular cyanobacterium Nostoc punctiforme. Environ Microbiol. 2014;16(3):829–44.

[Ramsay et al.] Ramsay, B., Wiedenheft, B., Allen, M., Gauss, G. H., Martin Lawrence, C., Young, M., and Douglas, T. (2006) Dps-like protein from the hyperthermophilic archaeon Pyrococcus furiosus. J. Inorg. Biochem. 100, 1061–1068

[Gauss et al.] Gauss, G.H.,Benas, P., Wiedenheft, B., Young, M., Douglas, T., and

Lawrence, C.M. (2006) Structure of the DPS-Like Protein from Sulfolobus solfataricus Reveals a Bacterioferritin-Like Dimetal Binding Site within a DPS-Like Dodecameric

Assembly Biochemistry, 45, 10815-10827

Table 1 – The authors should comment on why they did not process the data of Metal-free NpDps4 structure to a higher resolution based on their values of I/σ(I).

To be consistent the authors should present the values for highest resolution shell for the Rwork and Rfree.

-We present the original data from the ESRF synchrotron at high atomic resolution that fulfilled the aim of our study. The values Rwork and Rfree for highest resolution shell are now presented in the table 1.

Reviewer #2: The authors Howe et al., in their work about the fourth Dps from Nostoc punctiforme has tried to unravel its role in the organism and propose a new class of Dps, having His-type Ferroxidation centres unique to cyanobacteria. They also claim the NpDps4 uses O2 over H2O2 to oxidise iron. Also, crystal structures of apo, iron bound and zinc bound Dps has been analysed and they conclude by spectroscopic assays that binding of Zinc to the FOC inhibits ferroxidation.

While it’s interesting to note the presence of a new type of ferroxidation site in cyanobacterial Dps, the authors could not make convincing theories about its possible biological significance. There are also some major points that the authors need to address regarding their other findings to make their claims credible.

The following points are raised in the order they occur in the text:

Line 142

The residues Phe95, Leu98 and Ala99 are mentioned in the text but not marked in the figure.

-We have made a new figure showing these hydrophobic interactions, to be found in the supplementary data named ‘S1 Fig’.

Line 157

… acid of the trio …

-We rephrased accordingly.

Lines 168-169

“each symmetrical amino acid of the trio is indicated only once”, but Figure 3B shows amino acids from all three monomers related by three-fold.

-We revised the sentence accordingly.

Line 174

How does the authors know SOKALAN66 is a metal chelating agent?

-This information was provided by the manufacturer, BASF, and we have also experimentally observed that SOKALAN HP66 severely interferes with preparing metal-soaked crystals. Note further that SOKALAN HP66 is a vinylpyrrolidone/vinylimidazole copolymer, i.e there is an abundance of imidazole functional groups, a well-known metal ion chelator. We have included this information to the manuscript.

Line 176

Is there any experimental evidence whether it is really Zinc or Iron bound in the crystal structure like anomalous X-ray scattering or even a simple detection of Zinc ions on a gel?

-The anomalous maps have been included in the supplementary information (S2 Figure) confirming the presence of metal ions in these sites, however it is not possible to distinguish between Zn and Fe using these maps. Due to the high concentrations of metals in the soaking solution we are convinced that the strong electron densities found in the metal binding site indeed represent Fe and Zn respectively. The protein material, in which no metal could be found in the crystal, served as the basis for the soaking experiments. 

Line 177

It would be more desirable to show electron density maps to show the fitting of ion atoms in Figure 4

-Electron density is now shown in the new supplemental figure named ‘S2_Figure’.

Line 189

Since the FOC is formed not only of residues from two monomers related by two-fold, but also from a third monomer it is also unusual for Dps proteins.

-We agree, and as the results and discussion sections are separated in this manuscript, we had included this statement in the discussion section of the revised manuscript 

Lines 213-214

The sentence can be phrased better.

-We agree and we revised the sentence.

Line 220

What is the rationale of using 24 mM Fe2+ and 0.5 mM NpDps?

-At a ratio of 48 Fe2+/Dps protein all Fe2+ binding sites inside the Dps dodecamer could be theoretically occupied, as there are 12 FOC with each two iron binding sites per Dps protein (=24 Fe2+ / dodecamer binding sites). Now explained in the manuscript.

Lines 241-250

If the ferroxidation experiments using H2O2 were done under aerobic conditions, how does the authors discount for the effect of O2 on oxidation and especially as they’re comparing the ferroxidation properties between O2 and H2O2. It is strange that the graph doesn’t show any increase in ferroxidation under aerobic conditions, as in the above paragraph they show that oxygen can be used for ferroxidation by NpDps4.

-Yes we agree that this is a relevant comment. We did not discount for the effect of O2 on the reaction shown in Figure 5 B. It is likely that O2-mediated Fe2+ oxidation occurs in the presence of NpDps4, however this was not visible in the experiment. By using the same experimental set up we earlier observed a clear difference in the reaction between Fe2+ and H2O2 in the presence of NpDps1, NpDps2 and NpDps3 as compared to without protein (Howe et al. 2018). Thus showing a difference in Fe2+ oxidation in NpDps4 as compared to canonical Dps proteins.

It may be that NpDps4 utilises H2O2 for Fe2+ oxidation, but the catalytic effect has not been observed during the reaction as compared to the control in the absence of NpDps4. Changes in the text have been made accordingly.

Lines 262-263

Does the inhibition of ferroxidation by Zinc related to it causing protein aggregation as it is mentioned in line 263 that Zinc negatively impacts protein solubility?

-Currently we don’t see any relation between Zn2+ inhibition and this putative protein aggregation or precipitation process. 

Additionally we show that the addition of Zn2+ during the course of the O2-mediated Fe2+ reaction quickly inhibited NpDps4 (Fig. 6B), a much faster response as compared to the slow gradual increase of the background absorption. We revised some sentences in the result part for clarity.

Lines 282-288

Addition of Zinc indicated by arrow is misleading in Figure 6B as the black graph shows the reaction without Zn addition. In line 287 the authors mean figure 6B and not 6A?

-We agree with the reviewer that the arrow might be misleading in the Figure 6 B. To emphasize that no Zn2+ was added in the black kinetic trace. The figure was modified for clarity: “In the presence of 0.5 µM NpDps4 24 µM Fe2+ was added to the reaction mixture (black graph) and no Zn2+ was added during the course of the reaction”. Further clarifications were done to the figure. 

-Yes. It was meant 6 B, black graph, thank you!

Line 302

Figure 7 is illegible, and nothing can be inferred from it. So, the authors most important claim that the his-type FOC is conserved across cyanobacteria cannot be ascertained at all from the figure.

-The Figure was originally provided with 300 dpi. We provide the Figure now in 600 dpi.

Furthermore, we modified Figure 7 to highlight the variety of cyanobacteria, in which the His-type FOC was found. Figure 7 now contains further information about the strains’ morphology (unicellular or filamentous), their capacity of forming heterocysts for N2- fixation and their origin - whether isolated in a marine or freshwater environment [Uyeda et al.]. We hope that the figure now better illustrate that the his-type FOC is (not conserved but) widely spread among cyanobacteria. 

[Uyeda et al.]Uyeda, J. C., Harmon, L. J., and Blank, C. E. (2016) A comprehensive study of cyanobacterial morphological and ecological evolutionary dynamics through deep geologic time. PLoS One. 11, 1–32

Line 392

It’s not clear to the reader what is ‘the specific function’ in cyanobacteria.

-We agree that the sentence in L392 was not clear and we have now revised it.

Line 470

Recent developments on the dps-like interface and its role as an evolutionary switch between dps and ferritins has to be mentioned when discussing the role of this interface.

-The authors agree with reviewer #2 that the Dps-like interface displays a crucial structural element that evolutionarily is linked to maxiferritins. We included the following sentences at the end of the paragraph in the discussion section:

“However, the interface at the Dps-like pores has additionally been identified to be a crucial structural element that evolutionarily links maxiferritins with Dps proteins [56]. It was shown that a single mutation at this interface could render the dodecameric structure of a Dps protein into a maxiferritin during protein crystallization” 

Lines 620-621

During the ferroxidation experiment the authors talk of mixing the reaction by resuspension. How do they account for the reaction time that has elapsed during this process? Wherever possible cuvettes with stirrers should be used as these reaction rates can be pretty fast, especially the rates with H2O2.

-The spectroscopic data was collected to compare Fe-oxidation behaviour under different conditions and precise quantitative analysis was not the objective here. All the experiments have been conducted the same way and the time elapsed during mixing was between 8-10 sec long, for all experiments. By this experimental design we were able to determine qualitative differences. We did not attempt to extract kinetics and inhibitions parameters. Although very interesting, quantitation was not the aim of the ferroxidation analyses in this study.

Line 626

Ideally kinetic data should be evaluated from 3-4 experiments and at least two different protein preparations.

-The extraction of kinetic parameters for quantification was outside the scope of this manuscript and we performed two individual experiments for comparative qualitative analysis purposes (the replicate data is now included in the S1 file with raw data). For consistent we used the same protein material for both the crystallisation experiments and the spectroscopic experiment. Further experiments with a protein preparation of Strep-tagged NpDps4 showed very similar qualitative results (not included in this manuscript).

---

## [Editor Report · Decision Letter 1]

19 Jul 2019

The Dps4 from Nostoc punctiforme ATCC 29133 is a member of His-type FOC containing Dps protein class that can be broadly found among cyanobacteria

PONE-D-19-15019R1

Dear Dr. Stensjö,

We are pleased to inform you that your manuscript has been judged scientifically suitable for publication and will be formally accepted for publication once it complies with all outstanding technical requirements.

With kind regards,

Inês A. Cardoso Pereira, Ph.D.

Academic Editor

PLOS ONE
---

## [Editor Report · Acceptance letter]

24 Jul 2019

PONE-D-19-15019R1 

The Dps4 from *Nostoc punctiforme* ATCC 29133 is a member of His-type FOC containing Dps protein class that can be broadly found among cyanobacteria 

Dear Dr. Stensjö:

I am pleased to inform you that your manuscript has been deemed suitable for publication in PLOS ONE. Congratulations! Your manuscript is now with our production department. 

With kind regards,

on behalf of

Dr. Inês A. Cardoso Pereira 

Academic Editor

PLOS ONE